# Glia-mediated gut–brain cytokine signaling couples sleep to intestinal inflammatory responses induced by oxidative stress

Alina Malita, Anne H Skakkebaek, Olga Kubrak, Xiaokang Chen, Takashi Koyama, Elizabeth C Connolly, Nadja Ahrentloev, Ditte S Andersen, Michael J Texada, Kenneth Halberg, Kim Rewitz*

Department of Biology, University of Copenhagen, Copenhagen, Denmark

*For correspondence:
Kim.Rewitz@bio.ku.dk

## eLife Assessment

This **important** work by Malita et al. describes a mechanism by which an intestinal inflammatory response causes an increase in daytime sleep through signaling from the gut to the blood-brain barrier. Their findings suggest that cytokines Upd3 and Upd2 produced by the intestine following inflammation act on glia of the blood brain barrier to regulate sleep by modulating Allatostatin A signaling. The evidence is **compelling** and elegantly performed using the ample *Drosophila* genetic toolbox, making this work appealing for a broad group of neuroscience researchers interested in sleep and gut-brain interactions.

**Abstract** Sickness-induced sleep is a behavior conserved across species that promotes recovery from illness, yet the underlying mechanisms are poorly understood. Here, we show that interleukin-6-like cytokine signaling from the *Drosophila* gut to brain glial cells regulates sleep. Under healthy conditions, this pathway promotes wakefulness. However, elevated gut cytokine signaling in response to oxidative stress – triggered by immune and inflammatory responses in the intestine – induces sleep. The cytokines Unpaired 2 and –3 are upregulated by oxidative stress in entero-endocrine cells and activate JAK–STAT signaling in glial cells, including those of the blood–brain barrier (BBB). This activity maintains elevated sleep during oxidative-stress-induced intestinal disturbances, suggesting that the JAK–STAT pathway in glia inhibits wake-promoting signaling to facilitate sleep-dependent restoration under these conditions. We find that the enteric peptide Allatostatin A (AstA) enhances wakefulness, and during intestinal oxidative stress, gut-derived Unpaired 2/3 inhibits AstA receptor expression in BBB glia, thereby sustaining an elevated sleep state during gut inflammation or illness. Taken together, our work identifies a gut-to-glial communication pathway that couples sleep with intestinal homeostasis and disease, enhancing sleep during intestinal sickness, and contributing to our understanding of how sleep disturbances arise from gastrointestinal disturbances.

## Introduction

Sleep is a conserved behavior essential for physical health and mental well-being. This process maintains physiological balance and promotes recovery from illnesses and other stressors (*Imeri and Opp, 2009*; *Irwin, 2019*). In healthy states, animals exhibit rhythmic periods of wakefulness and activity, but their sleep significantly increases during illness. This adaptive behavior is related to the fundamental

**eLife digest** When we are sick, we often feel tired or sleepy. This sickness-induced sleep is a deeply conserved response across species that helps the body recover. While the immune system and the brain must somehow communicate to make this happen, we still know little about how signals from a sick body reach the brain to change sleep behavior.

The gut, for instance, plays an important role in health and illness, and inflammation in the gut is known to affect mental health and sleep. However, we do not fully understand how this inflammation might influence brain activity. To find out more, Malita et al. used the fruit fly *Drosophila* as a model to investigate how stress and inflammation in the gut might affect sleep, focusing on hormone-like signaling molecules called cytokines, which are involved in immune response and inflammation.

The researchers genetically engineered flies to eliminate the release of specific cytokines from the endocrine cells of the gut and tracked the animals' sleep and activity patterns. They next exposed flies to a chemical that triggers oxidative stress and inflammatory responses in the gut and monitored how this affected sleep. The flies were then dissected and stained for further immunohistochemical studies and confocal microscopy imaging.

The results revealed that oxidative stress triggers the release of specific cytokines from endocrine cells in the lining of the gut as part of an immune and inflammatory response. These cytokines travel through the body's circulatory system and activate a signaling pathway in glial cells that form the blood-brain barrier – the protective layer surrounding the brain. This pathway promotes sleep during intestinal stress and inflammation, likely to support recovery. Under healthy conditions, however, the same cytokine signals help keep the animal awake.

Malita et al. reveal a connection between the gut and the brain through which the intestine communicates its health status to the brain, enabling the animal to adjust its behaviors, such as sleep, in response to internal signals like inflammation or oxidative stress.

These findings help us understand how gut health influences sleep and mental well-being, and they may shed light on the sleep disturbances that often afflict people with gut disorders. While this work was done in fruit flies, the cytokine signaling pathways involved in disease exist in a similar form in humans. Further research is needed to determine whether similar gut-to-brain communication pathways that regulate sleep under conditions of intestinal illness exist in humans, which could eventually inform new strategies for managing sleep or mood disorders linked to gut inflammation.

role of sleep in the recovery process, allowing the body to conserve energy and allocate resources toward eliminating pathogens and repairing tissue damage. Sleep patterns are generated by neural circuits within the brain. These circuits engage in complex interactions involving diverse brain regions, neurotransmitters, and signaling pathways to regulate the cycles of sleep and wakefulness (*Eban-Rothschild et al., 2018*; *Shafer and Keene, 2021*). For sleep to be effectively modulated during illness, there must be a dynamic interaction between the physiological states of the body's organs and these central sleep-regulatory circuits. However, the signals that mediate this communication and the mechanisms by which they modulate sleep during health and disease remain poorly defined.

Sickness-induced sleep, a behavior conserved across species including mammals and flies (*Oikonomou and Prober, 2019*; *Toda et al., 2019*), is influenced by cytokines, which are key mediators of immune and inflammatory responses (*Imeri and Opp, 2009*; *Irwin, 2019*). Cytokines such as interleukin 1 (IL-1) and tumor necrosis factor alpha (TNFα) are expressed in the healthy mammalian brain in regions that are implicated in sleep regulation, and their circulating levels change during the normal sleep–wake cycle, peaking during the sleep phase. Furthermore, these factors' effects on sleep appear to be dose dependent, as low levels of IL-1 can enhance sleep, whereas higher doses can inhibit sleep, indicating a dual functionality. Since immune responses alter the expression of these cytokines, they have been hypothesized to act as 'somnogens' that promote sleep during times of infection or illness. However, the connection between sleep and immune function is bidirectional, since sleep deprivation in mammals has been linked to increased inflammatory response via IL-1 and TNFα (*Irwin, 2019*). In the fruit fly *Drosophila*, sleep deprivation also seems to influence TNFα in astrocyte-like cells to regulate homeostatic sleep responses that enable sleep rebound after deprivation (*Vanderheyden et al., 2018*). However, the effect of these cytokines on sleep has mostly

been linked to their central expression and function within the central nervous system (CNS), while the coupling of cytokines produced by peripheral tissues to sleep-regulatory systems within the brain remains poorly understood.

Disorders affecting the gastrointestinal tract can lead to sleep disturbances (*Marinelli et al., 2020*), which are also associated with virtually all mental illnesses (*Jagannath et al., 2013*; *Maurer et al., 2020*; *Winkelman and Lecea, 2020*). Conditions including depression, anxiety, and disturbed sleep are frequently observed in individuals with gut inflammation, and the gut microbiome has also been linked to sleep quality and mental health (*Marinelli et al., 2020*; *Hu et al., 2021*; *Li et al., 2018*; *Bisgaard et al., 2022*). These associations suggest a strong connection between gut health and sleep. For changes in gut status to bring about behavioral changes, the gut must sense its state of health, damage, or presence of pathogens and release signals that lead to altered cellular function within the brain. This gut-to-brain signaling is mediated in large part by hormonal factors released from specialized endocrine cells of the gut, the enteroendocrine cells (EECs) (*Lemaitre and Miguel-Aliaga, 2013*; *Latorre et al., 2016*). Like the mammalian intestine, the *Drosophila* gut produces numerous diverse hormones from specialized EECs (*Veenstra et al., 2008*; *Chen et al., 2016a*; *Hung et al., 2020*; *Guo et al., 2019*; *Koyama et al., 2020*; *Hung et al., 2020*). Some of these gut hormones are released in response to nutritional intake, and they diet-dependently modulate sleep patterns and arousability through communication with neuroendocrine centers and brain circuits (*Titos et al., 2023*; *Ahrentløv et al., 2025*; *Kubrak et al., 2022*).

In the fly, enteric infection or damage leads to the production of reactive oxygen species (ROS) and the increased expression of the IL-6-like inflammatory cytokines Unpaired 2 and –3 (Upd2/3) in the absorptive enterocytes, a response required for local gut regeneration (*Jiang et al., 2009*; *Buchon et al., 2013*). These cytokine factors signal through their receptor, Domeless (Dome), to activate the JAK/STAT signaling pathway in target cells, which is important for both immune function and metabolism in flies, demonstrating a conserved function of cytokine action in this species. While the three related cytokines Upd1, Upd2, and Upd3 all signal through Dome, Upd2 and in particular Upd3 are IL-6-like cytokines mainly triggered by infection and are directly linked with cellular immune responses (*Yang et al., 2015*; *Zandawala and Gera, 2024*; *Oldefest et al., 2013*). As in mammals, cytokines are also produced centrally within the *Drosophila* brain, and neuronal Upd1 acts in a leptin-like manner to regulate feeding (*Beshel et al., 2017*), a behavior that is also linked with sleep (*Shafer and Keene, 2021*). Peripheral cytokine signaling has also been shown to modulate sleep in this species, where the fat tissue releases Upd2 to reflect adequate nutrition, and this signal modulates sleep (*Ertekin et al., 2020*). Furthermore, Unpaired cytokines have been implicated in the modulation of feeding behavior through effects on glial cells (*Cai et al., 2021*). Glial cells, including those making up the blood–brain barrier (BBB), have recently gained attention for their role in sleep regulation in both flies and mammals (*Li et al., 2023a*; *Garofalo et al., 2020*; *Axelrod et al., 2023*; *Artiushin et al., 2018*). Neurons in the CNS are separated from the circulatory system by the BBB, a selectively semi-permeable cell layer (*Banks, 2008*), which presents a challenge for peripheral hormones to enter and signal to neurons within the brain. However, glial cells within the BBB are ideally positioned to receive and integrate systemic signals from peripheral organs and modulate neuronal function, thereby relaying peripheral information into the brain.

Here, we demonstrate Upd2 and Upd3 cytokine signaling from endocrine EECs in the intestine in *Drosophila*. Our findings show that Upd2/3 signaling from the EECs to BBB glial cells plays a dual role in sleep regulation. Under normal, healthy conditions, EEC-derived Unpaired signaling sustains wakefulness, whereas in response to oxidative stress that leads to gut inflammation, elevated Unpaired signaling instead promotes sleep. Stress-induced EEC-derived Upd2/3 activates the JAK–STAT pathway in glial cells at the blood–brain interface and adjusts sleep through this activation based on intestinal homeostasis and levels of inflammatory signaling from the gut. Our results suggest that gut-derived Unpaired signaling influences sleep regulation through glial gating of wake-promoting AstA-mediated signals, thus linking intestinal health with CNS-dependent behaviors. These results identify a gut–brain connection by which gut disease impacts sleep regulation.

## Results

### Gut-derived Unpaired cytokine signaling regulates sleep

To investigate whether cytokine signaling from the gut regulates sleep, we silenced the expression of *upd2* and *upd3* in the EECs, which are a principal endocrine cell type in the gut that releases signals with systemic effects and constitutes the functional basis of gut–brain signaling. Using *voilà-GAL4* (a driver that targets all EECs) to drive RNAi in EECs in conjunction with *Tub-GAL80^{ts}* (hereafter referred to as *voilà>*) for temperature-induced RNAi induction exclusively in the adult stage to prevent developmental effects, we observed significant knockdown of the main IL-6 cytokine, *upd3* (*Oldefest et al., 2013*), in dissected adult female midguts (*Figure 1A*), demonstrating that *upd3* is expressed in EECs under normal homeostatic conditions. To eliminate potential neuron-derived phenotypes, we employed *R57C10-GAL80*, a form of *nSyb-GAL80* that effectively inhibits neuronal GAL4 activity, in combination with *Tub-GAL80^{ts}* and *voilà-GAL4* (*Kubrak et al., 2022*; *Malita et al., 2022*). To evaluate the effectiveness of this temperature-sensitive EEC-specific driver (referred to as *EEC>* hereafter) upon adult-restricted induction, we examined midgut *upd2* and *upd3* transcript levels and observed significant knockdown of both cytokines, which was reproduced with a second independent RNAi line targeting *upd3* (*Figure 1B*). Importantly, we did not observe any effect of this manipulation on neuronal expression of *upd2* or *upd3*, supporting the specificity of EEC-targeted knockdown (*Figure 1—figure supplement 1A*). Additionally, *GFP* expressed under the control of an *upd3-GAL4* driver containing *upd3* enhancer sequences was apparent in midgut EECs, marked by staining against the EEC fate determinant Prospero (*Figure 1C*). To further confirm the expression of *upd3* in EECs, we conducted fluorescent in situ hybridization targeting *upd3* and *prospero* in adult midguts. This analysis revealed clear co-localization of *upd3* transcripts with *pros*-positive EECs, consistent with the expression pattern observed using *upd3-GAL4*-driven GFP in these cells (*Figure 1—figure supplement 1B*). These results show that EECs of the adult female midgut are a source of Upd2 and Upd3 under normal conditions.

We then explored whether these EEC-derived cytokines govern sleep under such normal conditions, given the known role of cytokines in both healthy and inflammatory states. We found that the knockdown of the two main immune-related cytokines, *upd2* or *upd3*, using *voilà>* increased the amount of time animals spent asleep (defined as a period of inactivity lasting at least 5 min), especially during the day (*Figure 1—figure supplement 1C–E*). This phenotype was reproducible using the more restricted *EEC>* driver that includes *R57C10-GAL80*, suggesting that the EEC-specific loss of these cytokines promotes sleep with pronounced effects during the daytime, when females are typically active (*Figure 1D–F*). This outcome was not attributable to off-target effects (two independent RNAi lines targeting the main IL-6-like cytokine, *upd3*, produced similar phenotypes) or to effects of the RNAi transgenes themselves (*Figure 1D–F*, *Figure 1—figure supplement 1F–H*). Animals lacking EEC-derived Unpaired signaling also exhibited shorter motion bouts (periods of activity; *Figure 1—figure supplement 1I, J*). However, the EEC-specific knockdown of *upd2* or *upd3* did not reduce motion-bout activity (the intensity of activity during wake periods), implying that the lack of gut cytokine signaling did not reduce general activity when the animals were awake (*Figure 1—figure supplement 1K, L*), and these effects do not arise from RNAi transgene insertions themselves (*Figure 1—figure supplement 1M, N*). These observations suggest a direct influence of gut-derived Upd2 and Upd3 on sleep rather than a broader impact on general activity levels. Given that sleep and feeding are mutually exclusive behaviors, we measured feeding. We did not detect significant alterations in feeding behavior as a consequence of *upd2* or *upd3* knockdown in EECs over a 24-hr period using the automated FLIC system (*Figure 1—figure supplement 1O*), nor did we observe an effect of EEC-specific *upd3* overexpression on food consumption via a dye assay (*Figure 1—figure supplement 1P*). Furthermore, EEC-specific *upd3* knockdown did not affect the animals' metabolic state as reflected in their levels of stored triacylglyceride (TAG) (*Figure 1—figure supplement 1Q*). Therefore, the sleep phenotype exhibited by animals with EEC-specific *upd2/3* knockdown is not associated with changes in metabolism, appetite, or feeding behavior.

To further explore the role of Upd2 and Upd3 in sleep regulation, we made use of deletion mutations that disrupt *upd3* alone (*upd3Δ*) or in combination with *upd2* (*upd2,3Δ*). Both mutant lines exhibited a pronounced increase in sleep (*Figure 1G–I*), with strong effects on daytime sleep, phenocopying the RNAi-mediated knockdown in the EECs. We additionally disrupted the *upd2* or *upd3* genes specifically in the EECs using somatic tissue-specific CRISPR-mediated deletion. EEC-specific

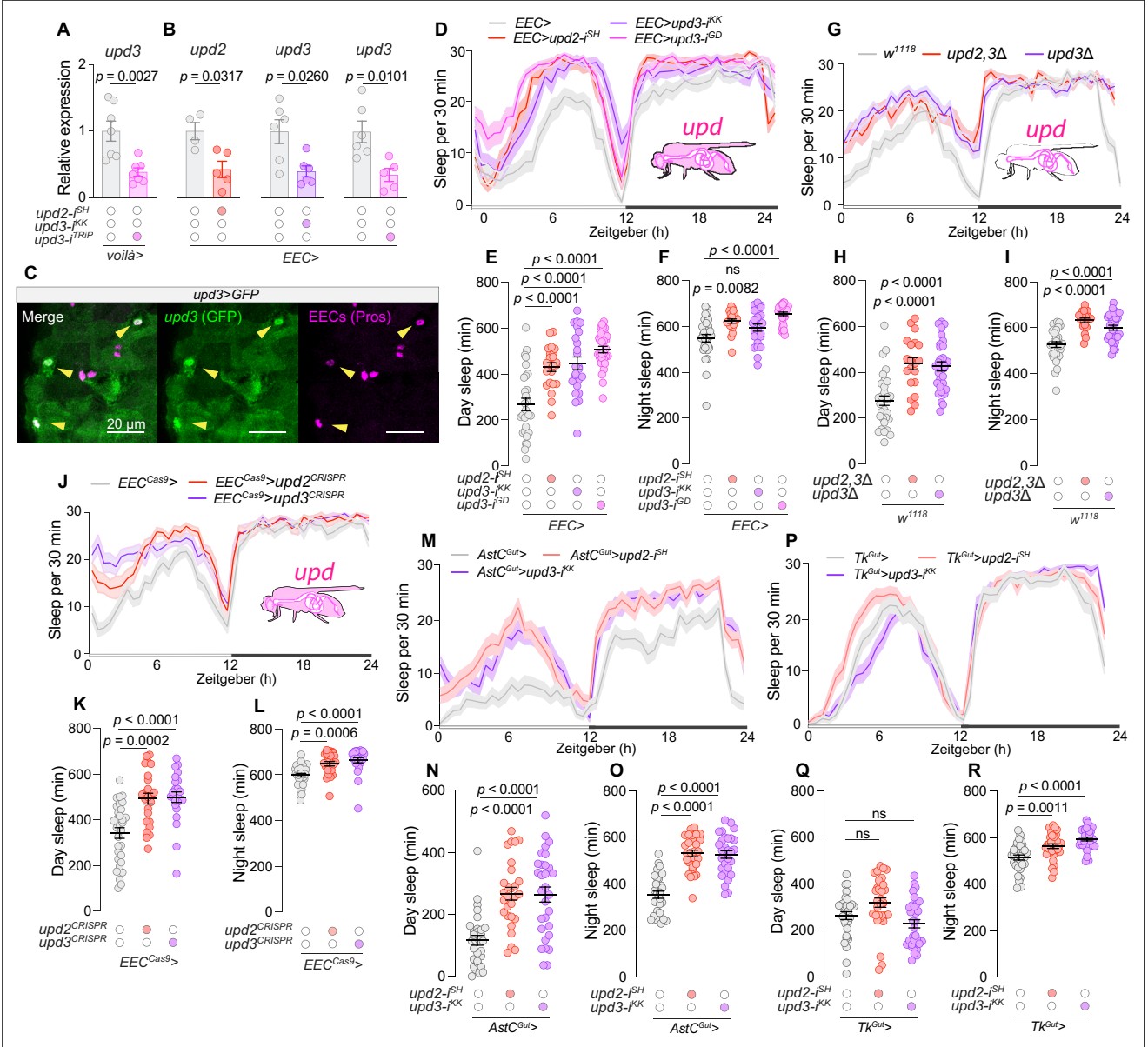

**Figure 1.** Enteroendocrine cell (EEC)-derived Unpaired signaling regulates sleep. (**A**) *upd3* expression levels in midguts expressing RNAi-mediated knockdown of *upd3* in EECs using *voilà-GAL4* (*voilà>*) in combination with *Tubulin-GAL80ts* (*voilà>*) (*N* = 7). (**B**) *upd2* and *upd3* expression levels in midguts in animals with EEC knockdown of *upd2* or *upd3* using *voilà-GAL4* in combination with *Tubulin-GAL80ts* and *R57C10-GAL80* (*EEC>*) (*N* = 4–6). (**C**), Confocal imaging of *upd3-GAL4*-driven *UAS-GFP* expression in EECs, co-stained with Prospero (Pros) as an EEC marker (scale bar, 20 μm). (**D**) Twenty-four-hour sleep profiles in animals with EEC-specific *upd2* or *upd3* knockdown (*N* = 29–32). (**E**) Total daytime sleep and (**F**) nighttime sleep durations in animals with EEC-specific *upd2*- or *upd3*-knockdown flies (*N* = 29–34). (**G**) Twenty-four-hour sleep profiles for global *upd2/3* deletion mutants (*N* = 19–32). (**H**) Daytime and (**I**) nighttime sleep durations in global *upd2/3* mutants (*N* = 19–32). (**J**) Sleep profiles following EEC-specific CRISPR-mediated *upd2* or *upd3* knockout (*N* = 24–31). (**K**) Daytime and (**L**) nighttime sleep durations in animals with EEC-specific CRISPR-mediated *upd2* or *upd3* knockout (*N* = 25–31). (**M**) Twenty-four-hour sleep profiles in animals with Allatostatin C (AstC)-positive-EEC-specific knockdown of *upd2* or *upd3* using *AstC-GAL4* combined with *R57C10-GAL80* (*AstCGut>*) (*N* = 29–30). (**N**), Daytime sleep, and (**O**) nighttime sleep durations in animals with *AstCGut>*-mediated knockdown of *upd2* or *upd3* (*N* = 28–32). (**P**) Twenty-four-hour sleep profiles in animals with Tachykinin (Tk)-positive-EEC-specific knockdown of *upd2* or *upd3* using *Tk-GAL4* combined with *R57C10-GAL80* (*TkGut>*) (*N* = 32). (**Q**) Daytime sleep and (**R**), nighttime sleep durations in animals with *TkGut>*-mediated knockdown of *upd2* or *upd3* (*N* = 28–31). Statistical analyses performed using Mann-Whitney tests for panels **A** and **B**; ordinary one-way ANOVA with Dunnett's multiple comparisons for panels **E**, **J**, **H**, **I**, **N**, **O**, **Q**, and **R**; Kruskal–Wallis ANOVA with Dunn's multiple comparisons for panels **F**, **K**, and **L**. Data are presented as mean ± SEM. ns, non-significant (p > 0.05). See also ***Source data 1***.

The online version of this article includes the following figure supplement(s) for figure 1:

**Figure supplement 1.** Effects on sleep, feeding, and metabolic parameters of Unpaired cytokine manipulation in enteroendocrine cells (EECs).

CRISPR-mediated knockout of *upd2* or *upd3*, induced by UAS-controlled gRNA pairs designed to excise portions of each gene's coding sequence, led to significantly elevated sleep (*Figure 1J–L*), further reinforcing these cytokines' role in sleep modulation.

To further dissect the cellular source of gut-derived cytokines regulating sleep, we analyzed the contribution of the two major EEC populations in the adult *Drosophila* midgut, marked by expression of either Allatostatin C (AstC) or Tachykinin (Tk) (*Guo et al., 2019*). These two molecularly defined groups encompass the vast majority of EECs. We used *AstC-GAL4* (*Kubrak et al., 2022*) and *Tk-GAL4* (*Ahrentløv et al., 2025*) drivers, which are knock-in lines carrying *GAL4* inserted at the endogenous *AstC* or *Tk* loci, thereby enabling precise genetic targeting of EECs based on their native hormone expression profile. To restrict GAL4 activity to the gut and thus avoid effects from neuronal expression, both drivers were combined with *R57C10-GAL80*, generating *AstC^Gut^-GAL4* (*AstC^Gut^>*) and *Tk^Gut^-GAL4* (*Tk^Gut^>*) drivers. Using these tools, we selectively knocked down *upd2* or *upd3* in either the AstC-positive EECs or the Tk-positive cells. Knockdown of either cytokine in AstC-positive EECs significantly increased sleep (*Figure 1M–O*), phenocopying the effect observed with knockdown in all EECs (*Figure 1D–F*). In contrast, knockdown of *upd2* or *upd3* in Tk-positive EECs had no significant effect on sleep (*Figure 1P–R*). These findings indicate that AstC-positive EECs are a major source of sleep-regulating Unpaired cytokines, whereas Tk-positive EECs do not appear to contribute significantly to this function. Consistent with this, we also observed effective knockdown of *upd3* transcripts in dissected midguts using the *AstC^Gut^>* driver, indicating that *upd3* is endogenously expressed in the AstC-positive EEC population (*Figure 1—figure supplement 1R*). Collectively, these findings demonstrate that Upd2 and Upd3 expressed by EECs are important modulators of diurnal sleep patterns in *Drosophila* under normal homeostatic conditions, and they further identify AstC-positive EECs as a key cellular source of these sleep-regulating cytokines.

## Glial cytokine JAK–STAT signaling regulates sleep

To identify the CNS targets of EEC-derived Upd2/3 cytokine signaling by which they regulate sleep, we examined the effects of targeted knockdown of the Upd2/3 JAK–STAT-linked receptor *dome* in neurons or glia, the two main cell types in the CNS. Pan-neuronal *dome* knockdown using the driver *R57C10-GAL4* (*Kubrak et al., 2022*) did not significantly alter daytime or nighttime sleep in adult females (*Figure 2A*), thus failing to recapitulate the sleep increase observed upon loss of *upd2* or *upd3* in EECs (*Figure 1*). This suggests that neurons are not the targets by which gut Unpaired cytokine signaling regulates sleep. In contrast, knockdown of *dome* in glial cells using the pan-glial driver *repo-GAL4* (*repo>*) resulted in a pronounced increase in daytime and nighttime sleep (*Figure 2A*), similar to the phenotype observed upon EEC-specific loss of Unpaired cytokines. To substantiate this observation further, we silenced *dome* expression in all glia using three independent RNAi lines, all of which strongly induced sleep during the day, effectively ruling out any off-target or transgene-background effects (*Figure 2B–D*, *Figure 2—figure supplement 1A–E*). Additionally, animals with glial *dome* knockdown displayed shorter daytime motion bouts, suggesting reduced periods of wakefulness (*Figure 2E*, *Figure 2—figure supplement 1F*), without exhibiting decreased activity during these bouts (*Figure 2F*, *Figure 2—figure supplement 1G*), phenocopying the effects of *upd2/3* knockdown in EECs (*Figure 1—figure supplement 1I, K*). We also assessed whether glia-specific *dome* knockdown might affect feeding and energy storage, but we observed no reduction in food intake (*Figure 2—figure supplement 1H*), and no changes in TAG levels were detected (*Figure 2—figure supplement 1I*). These findings collectively argue that augmented sleep resulting from impaired JAK–STAT signaling in glia is not due to a general decline in activity but rather represents a specific regulation of sleep itself.

To directly assess the functional Unpaired-mediated communication between the gut and glial cells, we manipulated *upd2* and *upd3* in EECs of animals carrying a ubiquitously expressed transgenic JAK–STAT reporter (*10xSTAT-GFP*) (*Bach et al., 2007*). Knockdown of either *upd2* or *upd3* in the EECs led to a marked decrease in JAK–STAT reporter activity within Repo-positive glial cells under normal conditions (*Figure 2G, H*), suggesting that Upd2/3-mediated signaling from the EECs to the brain's glial cells activates JAK–STAT signaling. Taken together, this suggests that gut-to-glia communication via Upd2 and Upd3 modulates diurnal sleep patterns through glial JAK–STAT activation and that these cytokines are required for the maintenance of wakefulness during the day under healthy conditions.

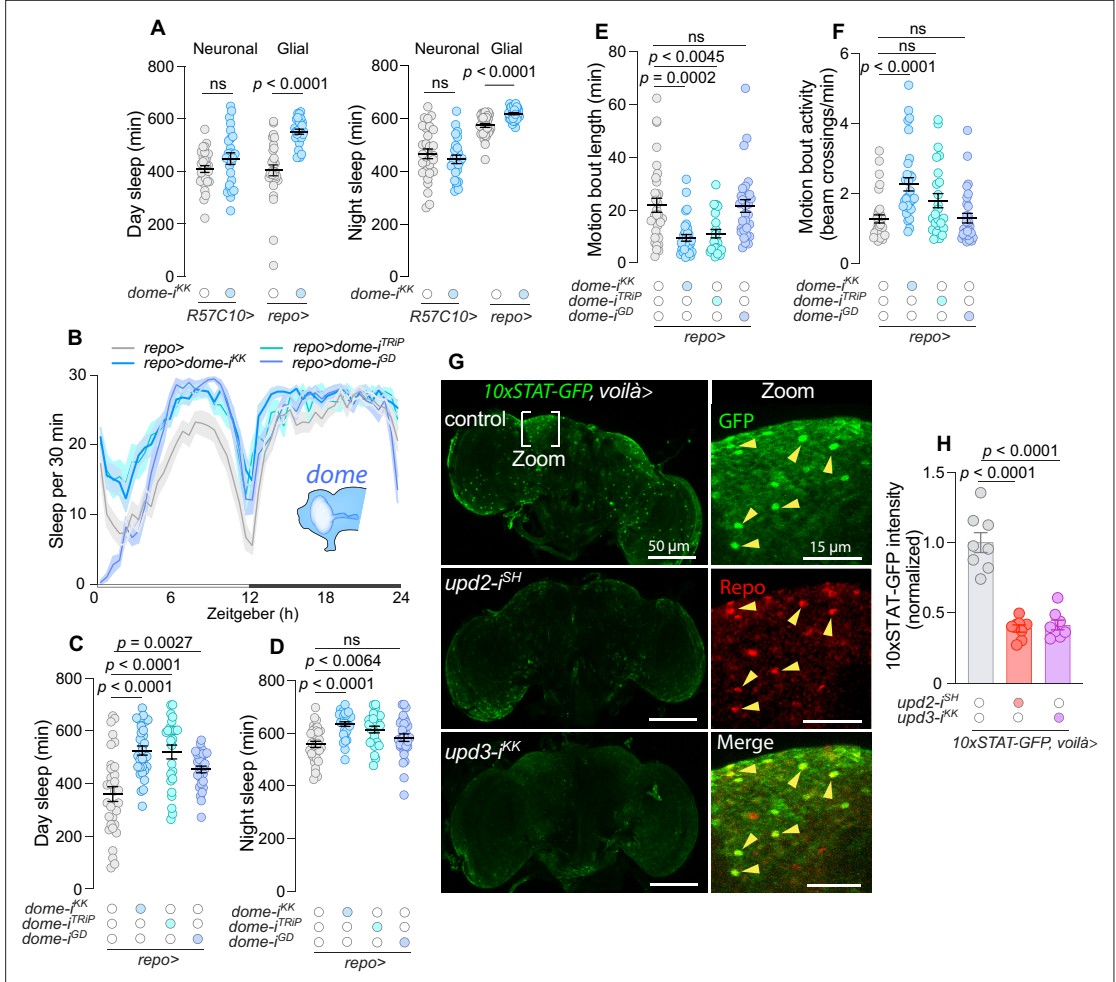

**Figure 2.** Enteroendocrine cell (EEC)-derived Unpaired signaling regulates glial JAK–STAT activity that modulates sleep. (**A**) Day and night sleep measurements for flies with knockdown of IL-6 related Unpaired cytokine receptor *domeless* (*dome*), which activates JAK–STAT, in neurons (*R57C10-GAL4, R57C10>*) and glial cells (*repo-GAL4, repo>*) (*N* = 25–32). (**B**) Twenty-four-hour sleep profiles for controls and animals with glia-specific *dome* knockdown (*N* = 25–32). (**C**) Total day sleep duration and (**D**) total night sleep duration for animals with glia-specific *dome* knockdown and control flies (*N* = 25–32). (**E**) Motion-bout length and (**F**) motion-bout activity in animals with glia-specific *dome* knockdown and controls (*N* = 25–32). (**G**) Representative images of brains from controls and animals with EEC knockdown *upd2* or *upd3* using *voilà>*, with *10xSTAT-GFP*. GFP expression (green) reflects JAK/STAT activity, and Repo labeling (red) indicates glial cells (scale bar, 50 μm). Insets show zoomed views of STAT-GFP+ and Repo+ glial cells (scale bar, 15 μm). (**H**) Quantitative analysis of GFP intensity in the layer of glial cells located at the surface of the brain in animals with EEC knockdown of *upd2* or *upd3* and controls (*N* = 8). Statistical analyses were conducted using parametric *t*-tests for panel **A**; Kruskal–Wallis ANOVA with Dunn's multiple comparisons for panels **C–F**; and ordinary one-way ANOVA with Dunnett's multiple comparisons for panel **H**. Data are presented as mean ± SEM. ns, non-significant (p > 0.05). See also *Source data 1*.

The online version of this article includes the following figure supplement(s) for figure 2:

**Figure supplement 1.** Characterization of sleep patterns, activity, and metabolic impact of *dome* knockdown in glial cells.

## Oxidative stress modulates sleep through gut-derived cytokine signaling

Having established the significance of gut-derived Unpaired cytokines in maintaining wakefulness under normal conditions, we next explored their role in sleep regulation in the context of gut disturbances that trigger immune and inflammatory responses. Gut infection leads to increased ROS levels and induces local cytokine production (*Jiang et al., 2009*; *Buchon et al., 2013*; *Lemaitre and Hoffmann, 2007*), and oxidative stress is a key feature of inflammatory conditions in the intestine (*Aviello and Knaus, 2017*). Dietary $H_2O_2$ treatment leads to local intestinal responses comparable to those observed during pathogenic challenges (*Tamamouna et al., 2021*), suggesting that $H_2O_2$ feeding

provides a controlled method to elevate intestinal ROS levels and examine the specific effects of ROS-induced cytokine signaling. We used this paradigm to ask whether intestinal oxidative stress might elevate the levels of Upd2 and Upd3 in the gut, and we found substantial upregulation of *upd3* expression in dissected midguts of females challenged with ROS by feeding 1% $H_2O_2$ in adult-specific, cornmeal-free diet for 20 hr (*Figure 3A*). This effect mirrored the upregulation observed with EEC-specific overexpression of *upd3*, indicating that it reflects physiologically relevant production of Upd3 by the gut in response to oxidative stress (*Figure 3A*). Oxidative stress also promoted *upd2* expression, albeit to a lesser extent, and this effect was not modulated by simultaneous EEC-specific *upd3* overexpression.

We next investigated whether sleep is modulated by intestinal oxidative stress and if Unpaired signaling from EECs is required for this response. We induced intestinal oxidative stress by exposing animals to a diet supplemented with $H_2O_2$ at Zeitgeber Time 0 (ZT0), the onset of the light phase, in a 12-hr light/dark cycle. Exposure to a lower $H_2O_2$ concentration (0.1%) incrementally increased daytime sleep amount over successive days (*Figure 3B*). In contrast, a higher $H_2O_2$ concentration (1%) triggered an immediate augmentation of daytime sleep (*Figure 3C*). Additionally, to ensure that the observed sleep increase was due to the presence of $H_2O_2$ itself rather than the procedure of food supplementation, we conducted a control experiment in which animals were fed standard food prepared using the same protocol and replaced daily, but without $H_2O_2$. These animals did not exhibit increased sleep, confirming that the effect is attributable to intestinal ROS (*Figure 3—figure supplement 1A*).

These observations suggest that intestinal oxidative stress dose-dependently modulates sleep. Since 1%-$H_2O_2$ feeding induced robust responses both in *upd3* expression and in sleep behavior, we asked whether gut-derived Unpaired signaling might be essential for the observed ROS-induced sleep modulation. Indeed, EEC-specific RNAi targeting *upd2* or *upd3* abolished the sleep response to 1%-$H_2O_2$ feeding. Animals with EEC-specific knockdown of *upd2* or *upd3* did not exhibit increased daytime sleep in response to the induction of oxidative stress in the intestine, even over two consecutive days of exposure to 1% $H_2O_2$-containing diet (*Figure 3D, E*). The specificity of this response was corroborated by three independent RNAi lines targeting *upd3*, negating the possibility of RNAi off-target effects (*Figure 3D, E*), and the loss of response to ROS was also not attributable to the transgenes themselves (*Figure 3—figure supplement 1B–E*). Intriguingly, animals lacking *upd3* in the EECs not only did not increase their sleep under oxidative stress but indeed appeared to lose nighttime sleep in response to enteric stress. Moreover, knockdown of *upd2* or *upd3* limited to the AstC-positive EEC subpopulation still prevented the $H_2O_2$-induced increase in sleep (*Figure 3F–H*). These findings indicate that Unpaired signaling from AstC-positive EECs is necessary for mediating the sleep response to intestinal oxidative stress and highlight a specific EEC subtype as a critical source of cytokine signaling in this context.

We next tested whether this sleep phenotype might be associated with general physiological processes rendering animals lacking EEC unpaired signaling more susceptible to ROS-induced damage. However, when we assessed survival on 1% $H_2O_2$-containing food, animals with *upd2* or *upd3* knockdown in EECs displayed no additional sensitivity to oxidative stress, compared to controls (*Figure 3I*). To further rule out nonspecific toxicity, we examined whether 1%-$H_2O_2$ feeding under our experimental conditions causes neuronal damage. Using a TUNEL assay for apoptosis, we found no evidence of increased neuronal cell death in animals fed 1% $H_2O_2$ for 24 hr, suggesting that the observed sleep phenotypes are not attributable to general neuronal toxicity (*Figure 3—figure supplement 1F and G*). This indicates that the loss of EEC-derived Unpaired signaling specifically leads to an impaired behavioral sleep response to intestinal oxidative stress, rather than to compromised physiological processes that would make the animals more vulnerable to oxidative-stress insults. We also examined whether animals lacking EEC-derived Unpaired signaling exhibit normal behavioral responses to other conditions that modulate sleep, which would suggest a specific requirement for this signaling in responding to intestinal oxidative stress. Animals typically suppress their sleep in response to nutritional deprivation, a behavior conserved across species that is believed to facilitate food-seeking activities and that is also influenced by EEC-mediated hormone signaling (*Kubrak et al., 2022*; *Lee and Park, 2004*). Animals lacking EEC-derived *upd3* suppressed their sleep to a similar extent as controls in response to starvation, indicating a normal sleep response to nutritional stress (*Figure 3—figure supplement 1H–J*).

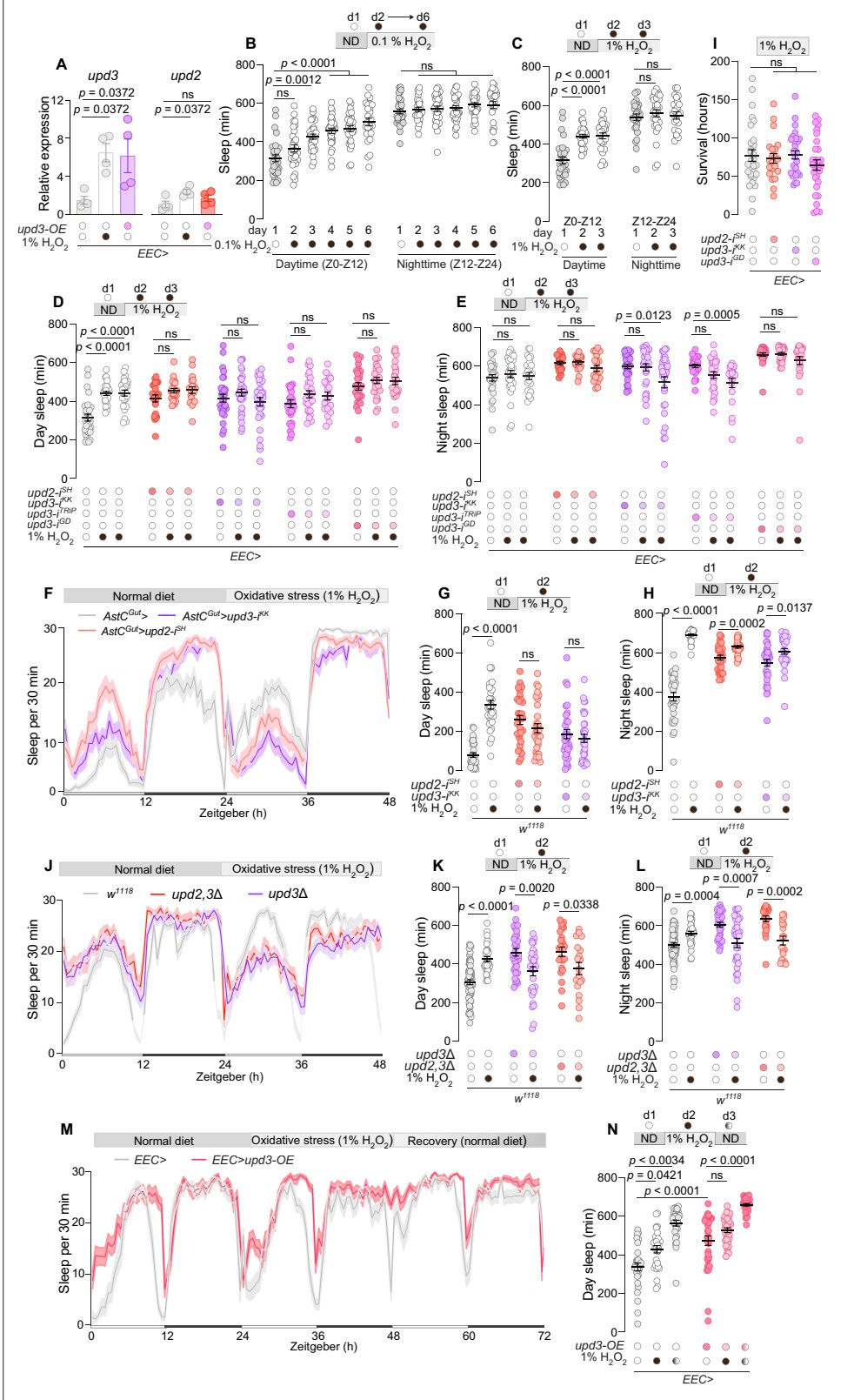

**Figure 3.** Enteroendocrine cell (EEC)-derived unpaired signaling modulates sleep in response to intestinal oxidative stress. (**A**) Measurement of *upd2* and *upd3* expression in the midgut upon 20 hr treatment with 1% $H_2O_2$-laced food or with overexpression of *upd3* (*upd3-OE*) using *EEC>* (N = 4). Assessment of sleep duration over consecutive days during the daytime (ZT0–ZT12) and nighttime (ZT12–ZT24) in animals exposed to food containing

*Figure 3 continued on next page*

*Figure 3 continued*

(**B**) 0.1% $H_2O_2$ (N = 26–31) or (**C**) 1% $H_2O_2$ (N = 23–29). (**D**) Daily sleep and (**E**) nightly sleep amounts measured over one day under standard food conditions followed by 2 consecutive days on 1% $H_2O_2$-containing food in animals with EEC-specific *upd2* or *upd3* knockdown and controls (N = 23–30). Experiments measuring sleep levels in controls and animals lacking EEC-derived *upd2* or *upd3* were performed concurrently and share the 'control' data, but results are presented in separate figures (B–E) for clarity. In (**D**) two-way ANOVA revealed significant genotype × diet interactions for *upd2-i* (p = 0.0076), *upd3-i^{KK}* (p = 0.0003), *upd3-i^{TRiP}* (p = 0.0204), and *upd3-i^{GD}* (p = 0.0040), relative to the control, indicating that the sleep response to oxidative stress depends on EEC-derived Unpaired signaling. (**F**) Sleep profiles and measurements of daytime (**G**) and nighttime (**H**) sleep across a 2-day period, encompassing 1 day on standard diet followed by 1 day on 1% $H_2O_2$-laced food to induce oxidative stress, in flies with AstC-positive-EEC-specific knockdown of *upd2* or *upd3* using *AstCG^{ut}>* compared to controls (N = 31–32). (**I**) Survival rates under a 1% $H_2O_2$-induced oxidative stress diet in controls and animals with EEC-specific *upd2* or *upd3* knockdown (N = 23–30). (**J**) A 48-hr sleep profile comparison between global *upd2/3* mutants and *w^{1118}* controls under 1 day of standard food conditions followed by 1 day of 1% $H_2O_2$-induced stress (N = 18–63). (**K, L**) Quantification of daytime and nighttime sleep durations in *upd2/3* mutants versus *w^{1118}* controls under normal-food conditions and the following day exposed to food containing 1% $H_2O_2$ (N = 18–63). In (**K**), two-way ANOVA showed significant genotype × diet interaction (p < 0.0001), confirming a role for Unpaired cytokines in reactive oxygen species (ROS)-induced sleep modulation. (**M**) Observation of sleep patterns, and (**N**) measurements of daytime sleep across a 3-day period, encompassing a day on standard diet, subsequent day on 1% $H_2O_2$-laced food to induce oxidative stress, and a final day back on standard diet to monitor recovery, in flies with EEC-specific overexpression of *upd3* (*upd3-OE*) compared to controls (N = 29–32). Statistical analyses were performed using Kruskal–Wallis ANOVA with Dunn's multiple comparisons for panels **A–E**, **I**, and **N**; Mann–Whitney test for panels **G**, **H**, **K**, and **L**. Interaction effects were assessed using two-way ANOVA where indicated. Data are presented as mean ± SEM. ns, non-significant (p > 0.05). See also *Source data 1*.

The online version of this article includes the following figure supplement(s) for figure 3:

**Figure supplement 1.** Sleep duration influenced by enteroendocrine cell (EEC)-specific unpaired cytokine disruption and dietary changes.

Although we observed behavioral phenotypes with manipulations of either *upd2* or *upd3* alone, suggesting that both are required for normal function, Upd2 and Upd3 likely function at least partially redundantly or additively in their regulation of sleep, as is the case for other processes (*Wang et al., 2014*). Moreover, RNAi effects do not result in a complete loss of function. Therefore, we speculated that a stronger disruption and combined knockout of both *upd2* and *upd3* might lead to even more pronounced phenotypes. We therefore tested the *upd3Δ* single-deletion line and the *upd2,3Δ* double-deletion mutants. Whereas *upd3Δ* and *upd2,3Δ* mutants exhibited increased baseline sleep under homeostatic conditions, these animals not only failed to increase their sleep in response to oxidative stress but indeed showed a strong reduction in daytime and nighttime sleep under oxidative-stress conditions (*Figure 3J–L*). These results suggest that, contrary to its role in promoting wakefulness during normal homeostatic conditions, the enhanced ROS-induced Unpaired signaling from EECs helps sustain a higher sleep level during periods of oxidative stress. This indicates a dual functionality of Unpaired cytokine signaling, in which low Unpaired signaling promotes wakefulness under normal conditions, whereas higher ROS-induced Unpaired signaling facilitates a shift to restorative sleep during intestinal stress.

We therefore investigated whether higher levels of Unpaired signaling from the gut, comparable to the level produced during oxidative stress, could enhance sleep in the absence of exogenous stressors. We analyzed the effect of *upd3* overexpression in EECs, which drives expression of midgut *upd3* to levels similar to those induced by 1%-$H_2O_2$ feeding (*Figure 3A*). Consistent with a model in which high levels of Upd3, like those that would occur during periods of elevated intestinal oxidative stress, promote daytime sleep, animals overexpressing *upd3* in the EECs exhibited increased sleep during the day, even in the absence of $H_2O_2$-induced oxidative stress (*Figure 3M, N*). These animals further increased their sleep in response to $H_2O_2$-induced enteric oxidative stress, unlike those lacking gut-derived *upd3* (*Figure 3D, E*), suggesting they remain able to mount an additional ROS-induced Unpaired signaling response on top of the overexpression-induced levels. After the animals were switched back to normal food after one day of oxidative stress, both control animals and those with EEC-specific *upd3* overexpression exhibited even more sleep than during the previous day under oxidative-stress conditions (*Figure 3M, N*). This suggests a robust recovery-sleep response following

the insult, likely mediated by Unpaired signaling, since the effect is more pronounced with *upd3* over-expression. Taken together, our results show that control animals increase their sleep during oxidative stress, likely as an adaptive recovery response. In contrast, animals with EEC-specific knockdown of *unpaired* cytokines do not exhibit this ROS-induced sleep response; instead, they experience sleep *loss* under such conditions. This suggests that while Unpaired signaling promotes wakefulness during normal healthy conditions, temporary ROS-induced elevation of gut Unpaired signaling suppresses arousal and leads to more sleep.

## EEC-derived Unpaired cytokine signaling activates glial JAK–STAT under oxidative stress

To investigate whether oxidative stress enhances glial JAK–STAT signaling and, if so, whether this enhancement might be mediated by gut-derived Upd2 and Upd3, we assessed glial JAK–STAT reporter activity using the dual-color TransTimer system, which provides temporal information about JAK–STAT signaling (*He et al., 2019*). In this system, active STAT promotes the expression of a construct encoding a short-lived destabilized GFP (dGFP, half-life ~2 hr) and a long-lived RFP (half-life ~20 hr) separated by a 2A peptide (*6xSTAT-dGFP:2A::RFP*); a higher ratio of GFP to RFP in a given cell reflects more recent JAK–STAT signaling. We explored whether JAK–STAT signaling responds dynamically to intestinal oxidative stress and assessed two daily time points. In control animals, we observed no circadian changes between ZT0 (lights on) and ZT12 (lights off) in the superficial layer of cells surrounding the brain (*Figure 4A, B*), which is composed of glia (*Freeman, 2015*). However, we observed a significant increase in GFP signal at ZT0 in animals fed for 20 hr with 1% $H_2O_2$-containing food, indicating recent JAK–STAT activity in the surface glia. Next, we investigated whether gut-derived Unpaired signaling is responsible for this upregulation by combining the *10xSTAT-GFP* reporter with knockdown of *upd2* or *upd3* in the EECs. Whereas glial JAK–STAT reporter activity was upregulated by oxidative stress (20 hrs' 1%-$H_2O_2$ feeding) in control animals, this response was abolished in animals with EEC-specific knockdown of *upd2* or *upd3*, indicating that this response is dependent on these EEC-derived cytokines (*Figure 4C, D*). Since in this case we used *voilà>* without the pan-neuronal *R57C10-GAL80* element to limit knockdown to EECs, we measured the expression of *upd2* and *upd3* in heads to check for any unintended neuronal effects that might contribute to the observed effect on glial JAK–STAT activity. We detected no changes in the expression of these genes in the head, confirming that the observed JAK–STAT activation in glial cells is attributable to cytokines derived from EECs (*Figure 4—figure supplement 1A, B*). To test the ability of gut-derived Upd3 to drive events in the brain in another way, we made use of cells' homeostatic response to changes in signaling input. Receptor expression is often upregulated in response to low levels of a ligand as a compensatory mechanism to enhance cellular sensitivity (*Puig and Tjian, 2005*). We observed an upregulation of *dome* transcript levels in the heads of animals with EEC-specific knockdown of *upd3*, exposed to oxidative stress induced by 15 hr of feeding with food laced with 1% $H_2O_2$ (*Figure 4E*). Increased *dome* expression suggests reduced Unpaired ligand availability as a result of the loss of EEC-derived Upd3. Together, our results demonstrate that EEC-derived Unpaired cytokine signaling is required for activating glial JAK–STAT under oxidative stress.

## Glial JAK–STAT modulates sleep in response to oxidative stress

Since EEC-derived Unpaired signaling promotes oxidative stress-induced sleep and glial JAK/STAT activity, we investigated whether the observed glial JAK–STAT signaling is involved in the modulation of sleep in response to intestinal oxidative stress. Knockdown of *dome* in all glial cells using the *repo>* driver completely abolished the ROS-induced daytime-sleep response when animals were fed a 1%-$H_2O_2$ supplemented diet. The specificity of this effect was confirmed using three RNAi lines and with transgenic RNAi controls (*Figure 5A*, *Figure 5—figure supplement 1A–C*). Mirroring the effects observed with *upd2,3Δ* mutants (*Figure 3G–I*), glia-specific *dome* knockdown (p < 0.00001 for *dome-i^KK* and p = 0.0556 for *dome-i^TRiP*) resulted in progressive and substantial sleep loss over two consecutive days on 1% $H_2O_2$-containing food. To rule out developmental effects, we restricted glial knockdown of *dome* to the adult stage using the *repo>* driver in combination with *Tub-GAL80^ts* (*repo^TS>*) and observed similar effects (*Figure 5B*).

We then examined the dynamics of sleep-regulatory glial JAK–STAT signaling by inducing oxidative stress for 1 day and then transferring the animals to normal food to observe the recovery response.

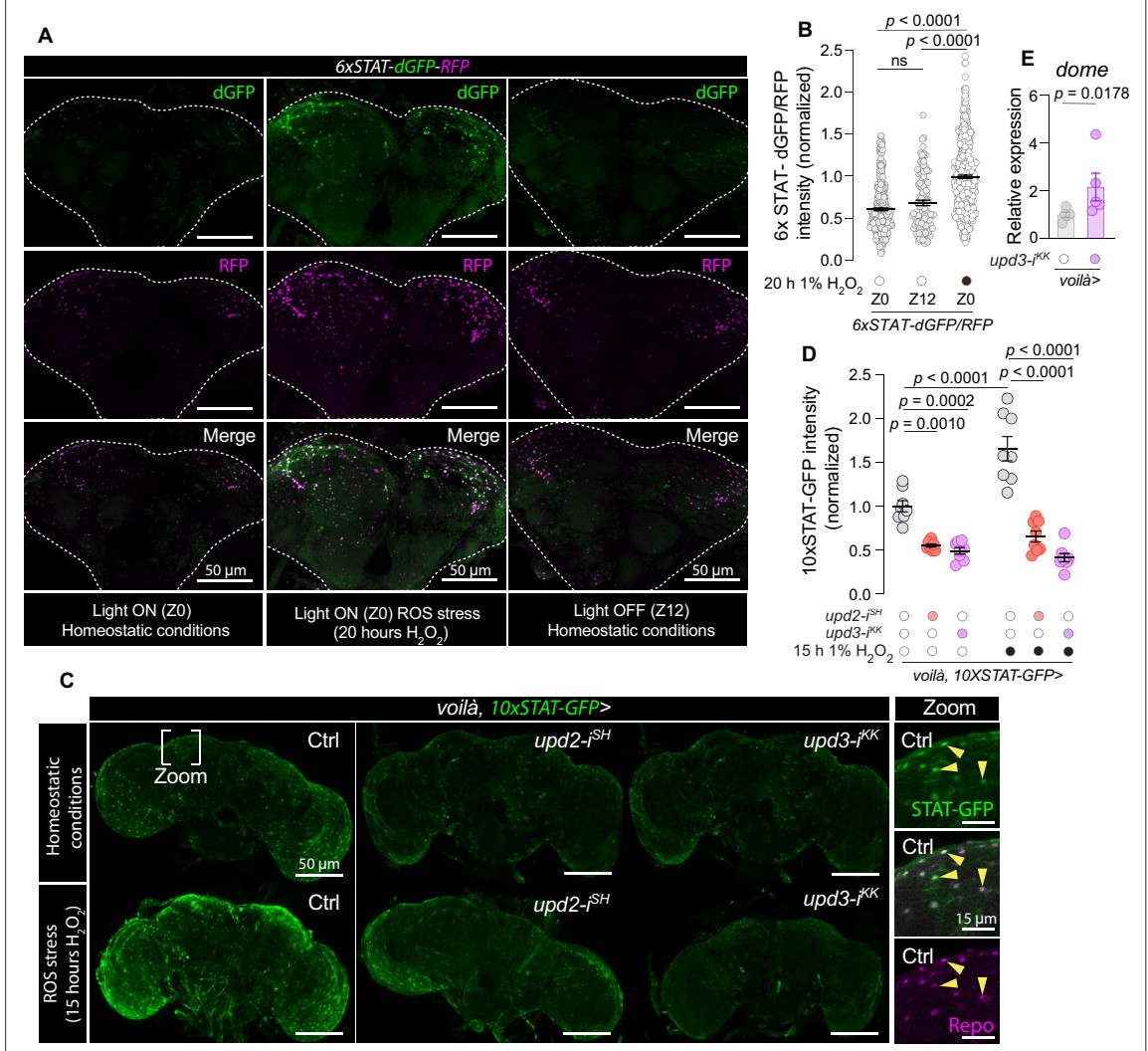

**Figure 4.** Activation of glial JAK–STAT signaling by enteroendocrine cell (EEC)-derived Unpaired cytokines in response to enteric oxidative stress. (**A**) Representative images of brains from flies expressing the *STAT-::dGFP::2A::RFP* reporter, where green (dGFP) reflects recent JAK–STAT activity due to its rapid degradation, and purple (RFP) indicates longer-term pathway activation due to its higher stability. Left panels show brains at lights-on (ZT0), middle panels show brains at lights off (ZT12), and right panels depict brains after 20 hr of oxidative stress induced by 1% $H_2O_2$-containing food, imaged at lights-on time ZT0. White dotted lines outline the brain perimeter. Scale bar, 50 μm. (**B**) Ratio of dGFP to RFP fluorescence intensity at ZT0, at ZT12, and after oxidative stress (at ZT0), as depicted in panel a, to show dynamic changes in JAK–STAT activity ($N$ = 108–461, indicating the number of cells counted). (**C**) Representative images of brains displaying *10xSTAT-GFP* expression under homeostatic conditions and after oxidative stress in control flies (Ctrl) and flies with EEC-specific *upd2* or *upd3* knockdown. Scale bar, 50 μm. Inset panels provide magnified views of glia cells labeled by anti-Repo. Scale bar, 15 μm. (**D**) Quantification of *10xSTAT*-driven GFP intensity in glial cells under homeostatic and oxidative-stress conditions, demonstrating the impact of EEC-specific cytokine knockdown ($N$ = 8, indicating the number of brains). (**E**), qPCR analysis of *dome* expression in the brains of flies with EEC-specific *upd3* knockdown in comparison to *voilà>* controls ($N$ = 5). Statistical analyses were conducted using Kruskal–Wallis ANOVA with Dunn's multiple comparisons for panel **B**; ordinary one-way ANOVA with Tukey's multiple comparisons for panel **D**; and two-sided unpaired *t*-tests for panel **E**. Data are presented as mean ± SEM. ns, non-significant (p > 0.05). See also *Source data 1*.

The online version of this article includes the following figure supplement(s) for figure 4:

**Figure supplement 1.** Expression of *upd2* and *upd3*.

The results showed that, in response to 1% $H_2O_2$-containing food, animals lacking glial *dome* expression displayed a sleep response opposite from that of controls, with a reduction in sleep duration rather than an increase, confirmed using independent RNAi lines (*Figure 5C*). This phenotype is similar to that seen in the *upd2,3Δ* double mutants (*Figure 3H–J*). During the recovery phase, after the animals had been switched back to normal food, the sleep level of controls increased even further, but the sleep duration exhibited by animals with glia-specific *dome* knockdown immediately reverted

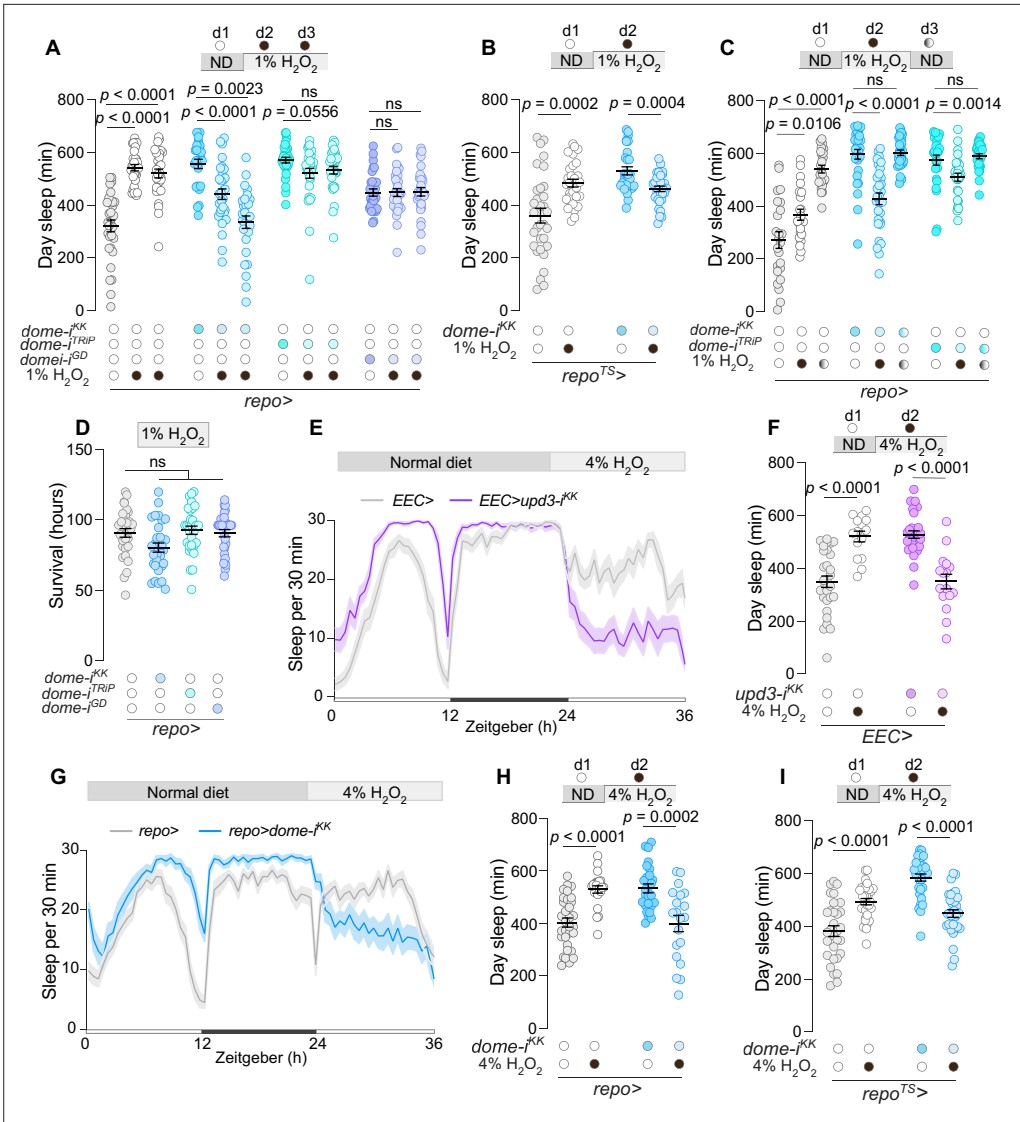

**Figure 5.** Enteroendocrine cell (EEC)-derived Unpaired and glial Domeless signaling modulate sleep during intestinal oxidative stress. (**A**) Daytime sleep duration in flies with glia-specific *dome* knockdown under standard and oxidative-stress conditions induced by 1% $H_2O_2$ in food (*N* = 25–32). Two-way ANOVA revealed significant genotype × diet interaction (p < 0.0001), indicating that glial Domeless is required for sleep regulation during oxidative stress. (**B**) Daytime sleep duration in flies with *repo*-driven *dome* knockdown restricted to the adult stage using *Tub-GAL80^ts* (*repo^TS*>) under normal conditions and during exposure to 1% $H_2O_2$-containing food (*N* = 29–32). Two-way ANOVA showed a significant genotype × diet interaction (p < 0.0001), further supporting a role for glial *dome* in regulating sleep in response to gut oxidative stress. (**C**) Daytime sleep during a 3-day period, encompassing a day on standard diet, subsequent day on 1% $H_2O_2$-laced food to induce oxidative stress, and a final day back on standard diet to monitor recovery, in controls and animals with glia-specific *dome* knockdown (*N* = 24–32). Two-way ANOVA revealed significant genotype × diet interaction (p < 0.0001). (**D**) Survival rates of controls and flies with glial-specific *dome* knockdown after exposure to oxidative stress by 1% $H_2O_2$-laced food (*N* = 31). (**E**) Sleep profiles and (**F**) daytime sleep duration for animals with EEC-specific *upd3* knockdown compared to control flies across a 36-hr period encompassing 24 hr on standard diet followed by 12 hr on oxidative-stress conditions induced by 4% $H_2O_2$-containing food (*N* = 15–30). Two-way ANOVA showed a significant genotype × diet interaction (p < 0.0001). (**G**) Sleep profiles and (**H**) daytime sleep duration for animals with glia-specific *dome* knockdown compared to control flies across a 36-hr period encompassing 24 hr on standard diet followed by 12 hr under oxidative-stress conditions induced by 4% $H_2O_2$-containing food (*N* = 20–32). Two-way ANOVA revealed significant genotype × diet interaction (p < 0.0001). (**I**) Nighttime sleep durations for animals under 4% $H_2O_2$ oxidative-stress conditions in controls and animals expressing adult-restricted knockdown of *dome* in glia (*N* = 31–

*Figure 5 continued on next page*

*Figure 5 continued*

32). Two-way ANOVA revealed significant genotype × diet interaction (p < 0.0001). Statistical tests: Kruskal–Wallis ANOVA with Dunn's multiple comparisons for panels **A**, **C**, and **D**; Unpaired two-sided *t*-tests for panels **B**, **F**, and **H**; and Mann–Whitney test for panel **I**. Interaction effects were assessed using two-way ANOVA where indicated. Data are presented as mean ± SEM. ns, non-significant (p > 0.05). See also *Source data 1*.

The online version of this article includes the following figure supplement(s) for figure 5:

**Figure supplement 1.** Impact of glia-specific *dome* knockdown on sleep patterns following oxidative stress and after sleep deprivation.

to pre-stress levels. This pattern indicates that animals with inhibited glial JAK–STAT signaling display an aberrant dynamic sleep response to oxidative stress that is not a consequence of a physiological breakdown but rather arises from altered inhibitory sleep-regulating mechanisms. In line with this and paralleling the loss of *upd2* or *upd3* in the EECs, *dome* knockdown in glial cells did not decrease survival on $H_2O_2$-containing food (*Figure 5D*). This supports the notion that physiological resistance to oxidative stress remains unaltered by gut-glial Unpaired signaling, which in turn indicates that the signaling modulation leads to a specific sleep phenotype. This is likely an important adaptive response under natural conditions, promoting recovery and maintaining homeostasis during or after transient stress episodes.

We further assessed whether glial loss of *dome* affected homeostatic sleep responses induced by sleep deprivation by evaluating the animals' ability to recover sleep after deprivation occurring during the second half of the night (ZT18–ZT24). Like controls, animals with glial-specific *dome* knockdown exhibited increased sleep (rebound sleep) in the morning hours (ZT0–ZT2) following sleep deprivation (*Figure 5—figure supplement 1D, E*). This indicates that they exhibit normal rebound sleep responses to deprivation and retain the capability to further increase their sleep. Collectively, these data suggest that Dome-mediated JAK–STAT signaling in the glial cells specifically regulates ROS-induced sleep responses.

We next investigated whether increased intestinal oxidative stress would exacerbate the phenotypes associated with the loss of *upd3* in EECs or *dome* in glial cells by exposing the animals to food containing 4% $H_2O_2$ and observing changes in their sleep architecture. Oxidative stress resulted in increased sleep in control animals, as anticipated (*Figure 5E–H*). However, in animals with EEC-specific *upd3* knockdown or glia-specific *dome* RNAi, exposure to 4% $H_2O_2$-containing food led to a pronounced loss of sleep during the daytime. For the EEC-specific *upd3* knockdown, the RNAi effect was induced at the adult stage (*Figure 5E, F*). We therefore also confirmed that adult-restricted knockdown of *dome* in glial cells resulted in similar phenotypes (*Figure 5I*). Thus, under conditions of intensified intestinal stress induced by 4% $H_2O_2$ in the food, the loss of *upd3* in EECs phenocopies the glial knockdown of *dome*, leading to reduced sleep and increased wakefulness.

## BBB glial JAK–STAT pathway activation drives sleep in response to intestinal oxidative stress

To determine the subset of glial cells responsible for mediating Unpaired-driven sleep regulation, we focused on the perineurial and subperineurial glial cells that form the BBB. These BBB glial cells serve as the interface between the CNS and the periphery, including its organs (*Freeman, 2015*), and are ideally situated to receive circulating signals from the intestine. Using the *10xSTAT-GFP* reporter, we assessed whether Upd3 from the gut activates JAK–STAT signaling within BBB glial cells. Knockdown of *upd3* in the EECs using *voilà>* (without *R57C10-GAL80*), which in this assay drives specific knockdown in the gut without detectable neuronal effects (*Figure 4—figure supplement 1A, B*), resulted in decreased GFP intensity in the outermost glial cell layer of the central brain facing the periphery after 15 hr of exposure to 1% $H_2O_2$-containing food, indicating reduced JAK–STAT activity in these cells (*Figure 6A, B*). Interestingly, this knockdown did not affect JAK–STAT activity in the outer glial layer of the ventral nerve cord (VNC), suggesting that Upd3 acts specifically on brain BBB glia in response to intestinal oxidative stress.

We further examined BBB-specific JAK–STAT signaling-mediated effects on sleep by specifically manipulating the subperineurial glial cells – those that form the permeability barrier – using *moody-GAL4* (*moody>*). As observed with other manipulations, knockdown of *dome* in these BBB glial cells led to increased sleep during normal homeostatic conditions (*Figure 6C–F*). Control animals

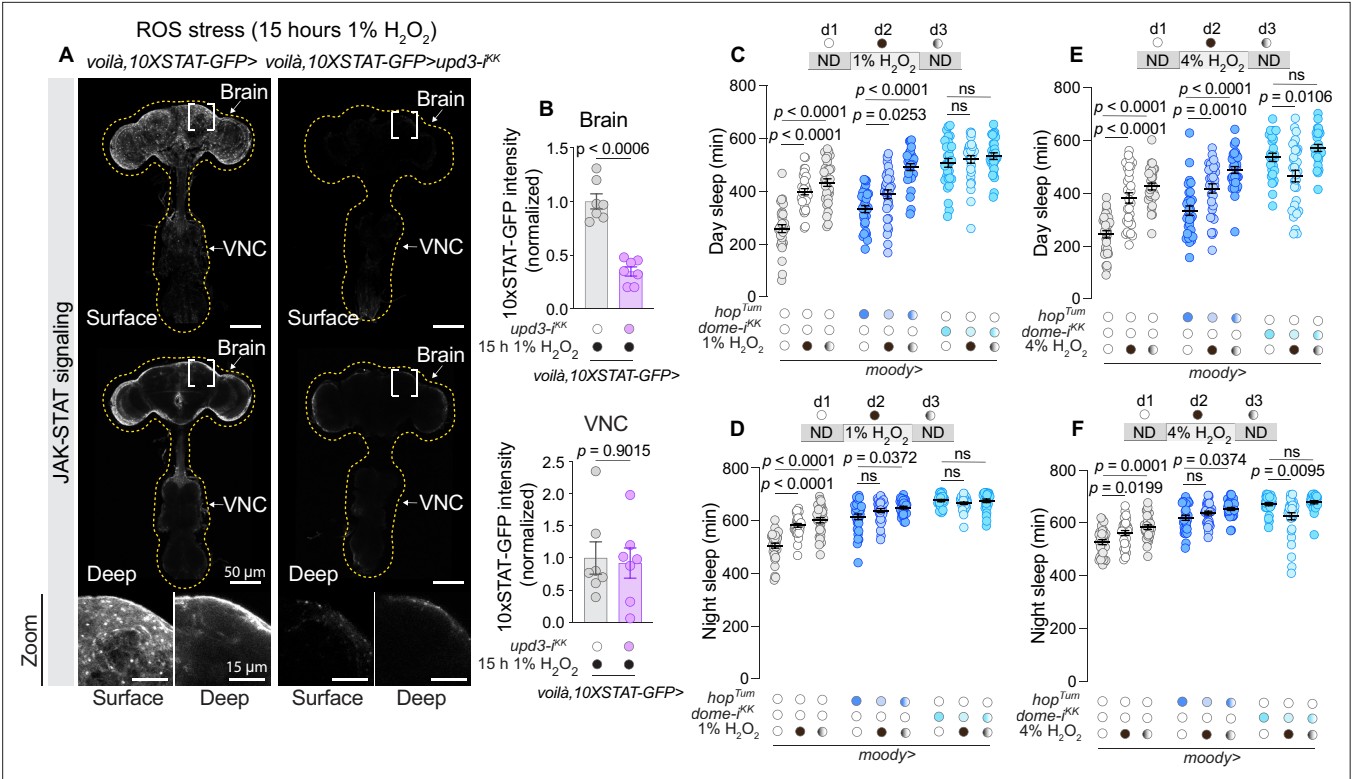

**Figure 6.** Blood–brain barrier (BBB) glia drive Domeless-mediated sleep responses to intestinal oxidative stress. (**A**) Representative images showing GFP expression driven by *10xSTAT-GFP* in controls and animals with *voilà-GAL4* (*voilà>*)-driven *upd3* knockdown in enteroendocrine cells (EECs) under intestinal reactive oxygen species (ROS) induced by 15 hrs' exposure to 1%-$H_2O_2$-containing food. The top panels depict overall brain and ventral nerve cord (VNC) structure with views of surface or deeper layers; the bottom panels provide zoomed-in views, highlighting the BBB glia at the interface between the brain and external environment. Dotted lines indicate brain and VNC perimeters. Scale bars, 50 µm (top) and 15 µm (bottom). (**B**) Quantification of GFP intensity in the brain and VNC in BBB glia in controls and animals with EEC knockdown of *upd3* under ROS stress, induced by exposure to 1% $H_2O_2$-laced food (*N* = 7). (**C**) Daytime and (**D**) nighttime sleep durations in flies with BBB-glia-specific knockdown of *dome* or overexpression of *hop^Tum^* under normal conditions, during exposure to 1% $H_2O_2$-containing food, and subsequent recovery on normal diet (*N* = 23–32). In (**C**) two-way ANOVA revealed significant genotype × diet interaction (p < 0.0001), indicating that BBB-glial Domeless is required for daytime sleep regulation under oxidative stress. (**E**) Daytime and (**F**) nighttime sleep durations in flies with BBB-glia-specific knockdown of *dome* or overexpression of *hop^Tum^* under normal conditions, during exposure to 4% $H_2O_2$-containing food, and subsequent recovery on normal diet (*N* = 23–32). In (**E**) two-way ANOVA revealed significant genotype × diet interaction (p < 0.0001), confirming the importance of BBB-glial Domeless signaling during higher levels of oxidative stress. Statistical tests used: Unpaired two-sided *t*-tests for panel **B**; Kruskal–Wallis ANOVA with Dunn's multiple comparisons in panels **C**, **D**, and **F**; ordinary one-way ANOVA with Dunnett's multiple comparisons for panel **E**. Interaction effects were assessed using two-way ANOVA where indicated. Data are presented as mean ± SEM. ns, non-significant (p > 0.05). See also ***Source data 1***.

exhibited the expected sleep increase both during 1%-$H_2O_2$ exposure and during the recovery period, but loss of *dome* in the subperineurial BBB glia blocked these effects (***Figure 6C, D***). When oxidative stress levels were elevated further using 4% $H_2O_2$-containing food, *dome-RNAi* in BBB glial cells led to sleep loss in response to oxidative stress, with sleep levels rebounding to pre-stress levels on the subsequent recovery day when animals were returned to a normal diet (***Figure 6E, F***). These results indicate that disrupting *dome* specifically in the subperineurial glial cells of the BBB recapitulates the phenotypes observed with pan-glial *dome* knockdown or with EEC-specific *upd3* knockdown, and they suggest that Dome-mediated JAK–STAT activation in the subperineurial BBB cells is required for maintaining an increased sleep state during intestinal oxidative stress.

To assess the sufficiency of subperineurial JAK–STAT signaling in inducing sleep, we activated the pathway in these cells by expressing a hyperactivated variant of the *Drosophila* JAK ortholog Hopscotch (Hop^Tum^). Expressing this protein in BBB glia led to increased sleep under normal conditions (without unusual oxidative stress), consistent with a sleep-promoting effect of high JAK–STAT signaling (***Figure 6C–F***). Moreover, animals with overactive JAK signaling in BBB glia exhibited a further increase in sleep both during oxidative stress and in the subsequent recovery phase, in contrast

to the effects seen with *dome* knockdown. This suggests that the combined activation of JAK–STAT induced by intestinal ROS and expression of $Hop^{Tum}$ leads to additive increases in sleep. Collectively, our data indicate that JAK–STAT signaling specifically in the subperineurial glial of the BBB links sleep responses to intestinal oxidative stress.

## AstA signaling promotes wakefulness and mediates ROS-induced sleep regulation in BBB glia

Our results indicate that the effect of gut-to-glia Unpaired cytokine signaling is both dose- and context-dependent. During intestinal oxidative stress, ROS-induced EEC Unpaired signaling leads to high JAK–STAT activity in subperineurial glial cells. Given that animals lacking this gut cytokine-to-glial signaling fail to maintain a high sleep state during oxidative stress, instead exhibiting increased wakefulness, this pathway appears to suppress wake-promoting signals under such conditions. The role of such wake-suppressive effects is likely to enhance sleep, aiding the process of recovery from intestinal damage. To identify potential wake-promoting signals that might be gated by JAK–STAT signaling, we examined a published dataset of genes whose expression in glia is positively or negatively correlated with these cells' JAK–STAT activity following enteric infection (*Cai et al., 2021*). The receptors for Allatostatin A (AstA), *AstA-R1* and *AstA-R2*, both ranked among the top 4% of genes most strongly downregulated by JAK/STAT signaling (with *AstA-R1* expression reduced by ~80% and *AstA-R2* by ~90%). Notably, these were the only peptide-hormone G-protein-coupled receptors downregulated in the JAK–STAT-activated glial cells. This suggests that upon intestinal infection, the JAK–STAT pathway is activated in glial cells, which suppresses AstA signaling by reducing the expression of the AstA receptors. Considering the central role of neuronal AstA in sleep-regulatory circuits (*Chen et al., 2016b*; *Dissel et al., 2022*), we investigated whether AstA might constitute a wake-promoting signal that is inhibited in glial cells by gut-derived Unpaired signaling. To evaluate the expression pattern of *AstA-R1* and *AstA-R2* within glial populations, we employed *AstA-R1-GAL4* and *AstA-R2-GAL4* knock-in constructs to drive the expression of nuclear-localized *RFP*. We co-stained the brain with antibodies against the glial transcription factor Repo, which marks the nuclei of glial cells. We observed that the outer layer of glial cells at the barrier between the brain and the periphery – constituting the BBB – expresses both *AstA-R1* and *AstA-R2* (*Figure 7A*). These findings are in line with previously reported data showing expression of *AstA-R1* and *AstA-R2* in glial cells referred to above (*Cai et al., 2021*). To functionally characterize the role of AstA signaling in these cells, we knocked down *AstA-R2* in BBB glia using *moody>*. This led to a significant reduction in *AstA-R2* transcript levels in dissected brains, indicating that BBB glia are a significant source of *AstA-R2* expression (*Figure 7—figure supplement 1A*). In support of an inhibitory role of Unpaired signaling, we observed that *AstA-R1* and *AstA-R2* expression was upregulated in the heads of animals with EEC-specific *upd3* knockdown, following gut-oxidative stress induced by feeding with 1% $H_2O_2$-laced food for 20 hr (*Figure 7B*). To demonstrate that this is caused by a failure to suppress AstA receptors in glial cells, we examined AstA receptor expression in brains following glia-specific *dome* knockdown in animals fed 1% $H_2O_2$-containing food for 20 hr. Indeed, glial-specific *dome* knockdown led to strong upregulation of both *AstA-R1* and *AstA-R2*, indicating that ROS-induced glia-mediated Unpaired signaling is inhibiting AstA receptor expression (*Figure 7C*).

Next, we investigated whether AstA receptors are involved in mediating the glia-regulated sleep response to intestinal oxidative stress. Like the Unpaired cytokines, AstA is released from EECs (*Chen et al., 2016b*) and may therefore act as a context-dependent wake-promoting signal that, under certain conditions, is inhibited by Unpaired signaling in BBB glia to promote sleep. We thus hypothesized that during intestinal disturbances characterized by oxidative stress, gut-derived unpaired signaling via JAK–STAT activation either sustains or consolidates sleep through a mechanism that involves the downregulation of wake-promoting AstA receptor signaling in BBB glial cells. In this model, EEC-derived Unpaired signaling normally suppresses AstA signaling in BBB glial cells under oxidative stress. Consequently, knocking down the *dome* in these cells or *unpaired* in the EECs leads to a failure to downregulate AstA receptors, causing the animals to wake up under these conditions. Thus, inhibition of glial AstA receptors would impair the animals' ability to respond to these wake-promoting signals altogether, leaving them unresponsive to intestinal ROS in terms of sleep. Consistent with this notion, we found that knocking down *AstA-R1* or *AstA-R2* in BBB glia attenuated the ROS-induced sleep response (*Figure 7—figure supplement 1B, C*). Knockdown of *AstA-R2* with a second,

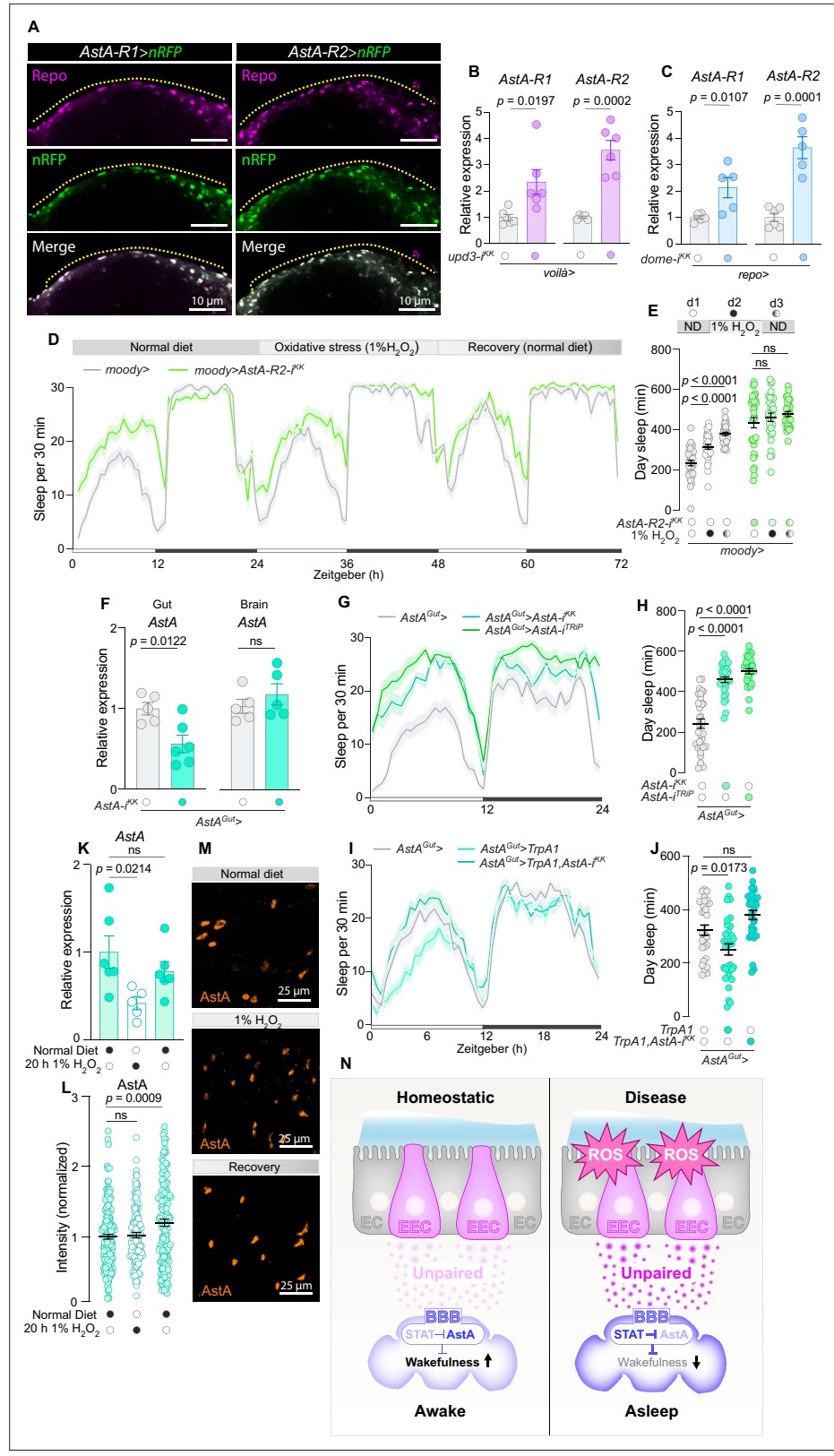

**Figure 7.** Gut unpaired cytokine signaling inhibits wake-promoting AstA signaling. (**A**) Images of brains from animals with *AstA-R1-GAL4* and *AstA-R2-GAL4* driving nuclear *dsRed* (nRFP, magenta) and co-stained with anti-Repo antibodies (green) show glial cells. The yellow dashed line indicates the interface between the brain and the external space, with the area below housing the blood–brain barrier (BBB) glial cells (scale bar, 10 μm). The

*Figure 7 continued on next page*

*Figure 7 continued*

yellow demarcation accentuates the separation between the cerebral interior and the external milieu, identifying the location of BBB glial cells underneath this partition (scale bar is 10 μm). (**B**) Relative expression of *AstA-R1* and *AstA-R2* in heads from animals with *upd3* knockdown in enteroendocrine cells (EECs) driven by *voilà-GAL4* (*voilà>*) compared to the control group after 20 hr on 1% $H_2O_2$-laced food to induce oxidative stress (*N* = 5–6). (**C**) Relative expression of *AstA-R1* and *AstA-R2* in brains from animals with *dome* knockdown in glial cells driven by *repo-GAL4* (*repo>*) compared to the control group after 20 hr on 1% $H_2O_2$-laced food to induce oxidative stress (*N* = 6). (**D**) Sleep patterns and (**E**) daytime sleep across a 3-day period, encompassing a day on a standard diet, a subsequent day on 1% $H_2O_2$-laced food to induce oxidative stress, and a final day back on the standard diet to monitor recovery, in flies with BBB-glia-specific knockdown of *AstA-R2* compared to control (*N* = 28–32). In (E), two-way ANOVA revealed significant genotype × diet interaction (p = 0.0114, supporting a role for glial AstA-R2 in reactive oxygen species [ROS]-induced sleep regulation). (**F**) *AstA* transcript levels in brains and midguts from controls and animals with knockdown of *AstA* in AstA$^+$ EECs using *AstA-GAL4* (*AstA>*) in combination with *R57C10-GAL80* to suppress neuronal GAL4 activity, referred to as *AstA$^{Gut}$>* (*N* = 5–6). (**G**) Sleep profiles and (**H**) daytime sleep on standard food of animals with *AstA* knockdown in AstA$^+$ EECs using *AstA$^{Gut}$>* and controls (*N* = 30–32). (**I**) Sleep profiles and (**J**) daytime sleep on standard food of controls, animals with TrpA1-mediated activation of AstA$^+$ EECs, and animals with TrpA1-mediated activation of AstA$^+$ EECs with simultaneous knockdown of *AstA* (*N* = 30–32). (**K–M**) Quantification of *AstA* transcript levels in whole midguts (**K**) and AstA peptide levels in the R5 region of the posterior midgut (**L**) on standard diet, after 1 day on 1% $H_2O_2$-laced food to induce oxidative stress, and during recovery following $H_2O_2$ exposure (k: *N* = 5–6; l: *N* = 126–170). (**M**) Representative images of R5 regions stained with anti-AstA antibody (scale bar: 25 μm). (**N**) Diagram illustrating the role of EECs in regulating wakefulness and sleep through Unpaired cytokine signaling under homeostatic and stress conditions. Left: Under homeostatic conditions, EECs release baseline levels of Unpaired, which interacts with the blood–brain barrier (BBB) to maintain normal JAK–STAT signaling and AstA transduction, promoting wakefulness. Right: In response to stress and disease, reactive oxygen species (ROS) increase in EECs, leading to elevated release of Unpaired. This surge in unpaired upregulates JAK–STAT signaling in BBB glia, which inhibits wake-promoting AstA signaling by suppressing AstA receptor expression, thus resulting in increased sleep, a state termed 'sickness sleep', to promote recovery. The diagrams depict the gut lining with EECs highlighted, the interface with the BBB, and the resulting systemic effects on the organism's sleep–wake states. EC: enterocyte. Statistical tests used: Unpaired two-sided *t*-tests for panel **B**, **C**, and **F**; ordinary ANOVA with Dunnett's multiple comparisons for panels **E**, **H**, **J**, **K**, and **L**. Interaction effects were assessed using two-way ANOVA where indicated. Data are presented as mean ± SEM. ns, non-significant (p > 0.05). See also ***Source data 1***.

The online version of this article includes the following figure supplement(s) for figure 7:

**Figure supplement 1.** AstA signaling from enteroendocrine cells (EECs) promotes wakefulness.

independent RNAi line resulted in a more pronounced phenotype, with an almost completely blunted sleep response to intestinal ROS, showing no significant sleep increase during oxidative stress or the following day of recovery (***Figure 7D, E***). AstA-R2 was also the more highly upregulated in response to loss of *upd3* in the EEC or *dome* in glia (***Figure 7B, C***), and it was the more strongly downregulated of the two AstA receptors in response to glial JAK–STAT activation, together suggesting that AstA-R2 is a primary receptor mediating these effects. Furthermore, knockdown of AstA receptors in BBB glia increased daytime sleep under normal homeostatic conditions, consistent with a wake-promoting role of AstA signaling in BBB glia (***Figure 7D, E***, ***Figure 7—figure supplement 1B, C***).

AstA is produced by two cell types, neurons and the EECs in the gut (***Chen et al., 2016b***). Since BBB glial cells are well-positioned to receive hormonal signals from the periphery, they likely are regulated by gut-derived AstA. We thus examined whether gut-derived AstA acts as a wake-promoting signal, by conducting *AstA* knockdown in AstA-positive EECs using an *AstA:2A::GAL4* knock-in in combination with *R57C10-GAL80* (*AstA$^{Gut}$>*) to suppress GAL4 activity in the AstA-positive neuronal population. We confirmed that this driver efficiently reduces the expression of *AstA* in midguts without affecting neuronal *AstA* transcript levels (***Figure 7F***). Knockdown of *AstA* in AstA-positive EECs with either of two independent RNAi constructs led to increased sleep without any contribution of the transgenic insertion backgrounds (***Figure 7G, H***, ***Figure 7—figure supplement 1D, E***), indicating that gut-derived AstA is indeed a wake-promoting factor. To assess whether EEC-derived AstA is sufficient to promote arousal, we employed the thermosensitive cation channel Transient Receptor Potential A1 (TrpA1) (***Hamada et al., 2008***) to induce hormonal release from AstA-positive EECs. Activation of these EECs suppressed sleep, an effect that was abolished by simultaneous *AstA* knockdown, supporting the wake-promoting role of EEC-derived AstA (***Figure 7I, J***).

Taken together, our findings suggest that enteric oxidative stress induces the release of Unpaired cytokines from the endocrine cells of the gut, which activate the JAK–STAT pathway in subperineurial glia of the BBB surrounding the brain. This activation leads to the glial downregulation of receptors for AstA, which is a wake-promoting factor also released by EECs. Gut-derived Upd signaling thereby gates the effect of AstA at the BBB and permits increased sleep during periods of intestinal stress. We therefore next investigated whether oxidative stress might also regulate the release of AstA from EECs. Following oxidative stress (24 hr of $H_2O_2$ feeding and the subsequent day), when wild-type animals exhibit increased sleep (*Figure 7E*), *AstA* transcript levels in the midgut were reduced, accompanied by an accumulation of AstA peptide (*Figure 7K–M*). This pattern – increased AstC staining in source cells despite decreased expression – suggests that oxidative stress suppresses AstA expression and release. This observation is consistent with a model in which, under conditions of enteric oxidative stress, wake-promoting gut-to-brain AstA signaling is silenced both at the source (gut EECs) and at the target (the BBB glia) by ROS-induced Unpaired signaling. We recently showed that Tk-positive EECs, which make up a population distinct from the AstA-positive EECs, express TrpA1, a ROS-sensitive cation channel known to promote hormone release, and thus exhibit ROS-induced Tk release (*Ahrentløv et al., 2025*). Contrasting with that system, single-cell RNA sequencing data (*Li et al., 2022*) show that the AstA-expressing EECs do *not* express TrpA1 (*Figure 7—figure supplement 1F*). This absence is consistent with our observation that oxidative stress does not promote AstA release (and indeed appears to inhibit it through mechanisms that remain to be explored), reinforcing the idea that gut-derived AstA signaling is actively suppressed rather than stimulated under these conditions. Together, these data support a model in which oxidative stress downregulates wake-promoting AstA signaling in the gut and simultaneously induces Unpaired cytokine signaling, which acts on BBB glia to suppress AstA receptor expression and thus to block the further transduction of wakefulness-promoting AstA signals. This dual-site regulation likely serves to silence arousal signals and promote sleep as a protective response to intestinal stress. This process may aid in recovery and maintain overall organismal homeostasis.

## Discussion

Intestinal inflammation and microbial imbalance are strongly associated with sleep disturbances and mental disorders such as anxiety and depression (*Marinelli et al., 2020*; *Hu et al., 2021*; *Li et al., 2018*; *Bisgaard et al., 2022*). The influence of gut health on CNS-dependent behaviors is thought to be mediated by the gut–brain axis, comprised of diverse signals secreted by the gut that act on the brain to induce behavioral responses (*Carabotti et al., 2015*). Whereas the regulation of feeding behavior by this axis has been extensively studied, leading to revolutionary approaches to medical weight loss and diabetes control, the role of gut–brain signaling in regulating sleep – a behavior affected across nearly all mental illnesses (*Glickman, 2010*; *Cohrs, 2008*) – remains poorly defined. Sickness induces a state of sleepiness, which is believed to be a conserved adaptive response that promotes recovery by supporting energy conservation and efficient immune activity (*Oikonomou and Prober, 2019*; *Toda et al., 2019*). However, the exact mechanisms driving sickness-induced sleep remain largely elusive. We have demonstrated here that intestinal ROS stress, through driving the release of interleukin 6-like Unpaired cytokines from endocrine cells of the *Drosophila* gut, regulates sleep via a glia-mediated pathway. This gut-to-glia communication promotes sleep during intestinal insult, presumably to facilitate the restorative sleep essential for both physical and mental health. Our findings provide mechanistic insight into how perturbations of gut health can influence sleep, potentially contributing to understanding the link between gastrointestinal disorders, sleep disturbances, and mental illnesses.

Cytokines, key secreted mediators of immune and inflammatory responses, are thought to modulate sleep/wake cycles under disease conditions (*Ditmer et al., 2021*). Interleukins and TNFα, cytokines induced during illness in mammals, have been suggested to promote sleep to aid recovery from disease. However, most of these effects have been attributed to the actions of cytokines produced within the CNS, leaving open the question of how diseases affecting other parts of the body can drive sleep responses. In *Drosophila*, sleep induced by immune responses is known to be influenced by the NFκB ortholog Relish in fat tissue (*Kuo et al., 2010*), and the neuronally expressed gene *nemuri* drives sleep and connects immune function with sleep regulation (*Toda et al., 2019*). However, inter-organ signaling mechanisms by which intestinal disease or stress regulate sleep have not yet been described

in either flies or mammals. Intestinal infection or inflammation leads to elevated levels of ROS in the gut, and our findings demonstrate that enteric oxidative stress in the gut triggers the production of Upd2 and Upd3 cytokines by hormone-secreting EECs. These gut-derived cytokines signal the state of the intestine to brain glial cells, including those of the BBB, and modulate sleep. This glia-mediated gut-to-brain signaling promotes wakefulness in healthy animals under normal conditions, while inducing sleep in response to oxidative stress in the intestine. This indicates a dual functionality, with low levels of gut Unpaired signaling promoting wakefulness and higher stress-induced levels acting to enhance sleep (*Figure 7N*). A similar dose-dependent effect has previously been observed for interleukins in rats, in which injection of IL-1 into the CNS can either stimulate sleep or inhibit it, depending on the administered dose (*Opp et al., 1991*). While our findings show that ROS-induced cytokine signaling in the gut modulates sleep through gut–brain communication, an intriguing direction for future research will be to determine whether pathogenic infections – which trigger both intestinal ROS and additional immune pathways – engage distinct, complementary, or overlapping mechanisms compared to chemically induced oxidative stress, and how these immune responses collectively influence sleep regulation.

Our results indicate that Unpaired signaling in subperineurial glial cells – those forming the BBB – activates the JAK–STAT pathway, and they suggest that this effect inhibits wake-promoting AstA signaling by downregulating AstA receptor expression. AstA and its receptors, which are orthologous with the mammalian Galanin signaling system, have been linked to the regulation of sleep, feeding, and metabolism (*Chen et al., 2016b*; *Hentze et al., 2015*; *Hergarden et al., 2012*). Mammalian glia express receptors for Galanin (*Priller, 1998*), which also regulates sleep (*Ma et al., 2019*; *Reichert et al., 2019*), further underscoring a conserved role in sleep modulation across species. AstA-producing neurons induce sleep by releasing glutamate onto sleep-regulatory neuronal circuits, although recent findings also suggest a wake-promoting role for AstA signaling (*Dissel et al., 2022*). Irrespective of neuronal AstA, our experiments clearly show that AstA released from EECs of the gut acts as a wake-promoting signal and that activation of AstA receptor signaling in BBB glial cells induces wakefulness. This highlights the potential of peptide hormones to elicit different effects depending on their source tissue and thus their accessible target cells – whether they are produced by the gut outside the BBB or by the CNS inside the barrier. A similar phenomenon has been demonstrated for neuropeptide F (*Malita et al., 2022*; *Chung et al., 2017*). Our findings further suggest that AstA release from EECs is downregulated under oxidative stress in the gut, indicating that this wake-promoting signal is suppressed both at the level of the intestine and at the BBB via Unpaired cytokine signaling. This coordinated downregulation may serve to effectively silence this arousal pathway and promote sleep during intestinal stress.

Interestingly, intestinal ROS can also be generated as a consequence of sleep deprivation (*Li et al., 2023b*; *Vaccaro et al., 2020*), suggesting a potential feedback mechanism. This raises the possibility that ROS produced during sleep loss engages the same Unpaired–JAK–STAT signaling cascade described here, leading to suppression of gut-derived AstA signaling and facilitating recovery sleep. This model provides a mechanistic link between sleep deprivation, intestinal stress, and the regulation of sleep and suggests that ROS may serve as a physiological signal integrating peripheral stress and behavioral state.

While our study investigated the effects of ROS induction, contrasting findings have been reported under conditions of antioxidant treatment (*Li et al., 2023b*). Our data show both decreased *AstA* transcript levels and increased AstA peptide accumulation following oxidative stress – a combination typically interpreted as reduced production coupled with peptide retention (*Ahrentløv et al., 2025*; *Kubrak et al., 2022*; *Malita et al., 2022*). In contrast, the reported increase in AstA peptide levels under antioxidant treatment was not accompanied by expression data (*Li et al., 2023b*), making it difficult to determine whether the AstA accumulation under these conditions reflects enhanced retention and/or increased production. Furthermore, single-cell RNA sequencing data (*Li et al., 2022*) indicate that AstA-positive EECs do not express the ROS-sensitive cation channel TrpA1, supporting our observation that intestinal ROS does not stimulate AstA release. We recently found that TrpA1 is expressed in a distinct population of Tk-positive EECs and drives ROS-dependent release of the gut hormone Tk from these cells in *Drosophila*. This mechanism was also observed in the mammalian intestine (*Ahrentløv et al., 2025*). In contrast, a previous report suggested TrpA1-dependent AstA release from EECs (*Li et al., 2023b*), highlighting a potential discrepancy in whether this channel

regulates AstA secretion. These differences may reflect context-specific variation in enteroendocrine function, and in any case, they underscore the complexity of AstA regulation under varying conditions of gut stress.

*Drosophila* exhibit conserved behaviors such as sleep, arousal/wakefulness, and anxiety-like responses (*Shafer and Keene, 2021*; *Yuan et al., 2006*; *Johnson et al., 2009*; *Mohammad et al., 2016*; *Gibson et al., 2015*; *Sehgal, 2017*; *Shaw, 2003*), and the EECs of the fly gut produce diverse hormones similar to those of mammals (*Veenstra et al., 2008*; *Chen et al., 2016a*; *Hung et al., 2020*; *Guo et al., 2019*; *Koyama et al., 2020*; *Hung et al., 2020*), potentially influenced by diet, microbiota, and inflammatory responses. This makes *Drosophila* an excellent model for studying behaviors influenced by gut conditions through gut–brain signaling. Our findings suggest that the oxidative-stress level within gut tissues, which is modulated by intestinal bacteria and immune activity (*Jiang et al., 2009*; *Buchon et al., 2013*), regulates sleep via EEC-derived Unpaired signaling, potentially explaining the observed links between gut microbiota and sleep disturbances in both flies and humans (*Li et al., 2018*; *Silva et al., 2021*). Furthermore, in mammals, conditions such as inflammatory bowel disease that are linked with oxidative stress (*Rezaie et al., 2007*) are often associated with sleep and mental health disturbances (*Marinelli et al., 2020*; *Hu et al., 2021*; *Bisgaard et al., 2022*). Our results imply that cytokines, including interleukin signaling from an inflamed or diseased gut, might be a mechanism by which intestinal illnesses affect sleep and mental health. Our findings raise the possibility that these cytokines may act on glial cells that integrate and relay these gut signals to brain sleep-regulatory circuits.

The neurons of the CNS are isolated from the circulatory system by the BBB (*Yildirim et al., 2019*) that restricts the transmission of some hormonal and cytokine signals from the periphery to neurons within the brain. Our work suggests that the BBB receives AstA and Unpaired signaling from the periphery. Other reports indicate that Unpaired cytokines from tumors and from enterocytes also can activate JAK–STAT signaling in BBB glia cells in *Drosophila* (*Cai et al., 2021*; *Kim et al., 2021*). Although our findings highlight endocrine EECs as a primary source of gut-derived cytokines that act on the brain to regulate sleep, it is also possible that enterocytes or other non-endocrine gut cell types contribute to the systemic Unpaired signaling that modulates sleep in response to intestinal oxidative stress. One effect of glial JAK–STAT activity seems to be the alteration of BBB permeability (*Kim et al., 2021*), raising the possibility that EEC-derived Unpaired signaling in BBB glia, directly or through AstA signaling, modulates sleep via regulation of BBB permeability, which has been linked to homeostatic sleep regulation (*Axelrod et al., 2023*). Furthermore, the endocytic activity of BBB glia, important for cellular transport and barrier function, has also been associated with sleep regulation (*Artiushin et al., 2018*), and thus JAK–STAT-induced changes could regulate sleep through alterations in intracellular trafficking within the cells of the BBB. Another possibility is that JAK–STAT activity might regulate glial metabolic support for neuronal activity and in this way affect sleep patterns. In any case, our findings highlight the involvement of BBB glial cells in transmitting signals from the gut to the brain, adding another layer to our understanding of body-to-brain communication, which suggests that the BBB does more than protect the brain; it also responds to peripheral signals to modulate brain function, presenting an intriguing area for future research into gut–brain signaling.

# Methods

## Key resources table

| Reagent type (species) or resource | Designation | Source or reference | Identifiers | Additional information |
|---|---|---|---|---|
| Genetic reagent (*Drosophila melanogaster*) | 10xSTAT-GFP | Gift of Julien Colombani | | |
| Genetic reagent (*D. melanogaster*) | 6xSTAT-dGFP::2A::RFP | Gift of Norbert Perrimon | | |
| Genetic reagent (*D. melanogaster*) | AstA::2A::GAL4 | University of Indiana Bloomington *Drosophila* Stock Center (BDSC) #84593 | RRID:BDSC_84593 | |
| Genetic reagent (*D. melanogaster*) | AstA-R1::2A::GAL4 | BDSC #84709 | RRID:BDSC_84709 | |

*Continued on next page*

*Continued*

| Reagent type (species) or resource | Designation | Source or reference | Identifiers | Additional information |
|---|---|---|---|---|
| Genetic reagent (*D. melanogaster*) | *AstA-R2::2A::GAL4* | BDSC #84594 | RRID:BDSC_84594 | |
| Genetic reagent (*D. melanogaster*) | *AstC::2A::GAL4* | BDSC #84595 | RRID:BDSC_84595 | |
| Genetic reagent (*D. melanogaster*) | *AstC^{Gut}>* | https://doi.org/10.1038/s41467-022-28268-x | | *R57C10-GAL80-WPRE; AstC::2A::GAL4* |
| Genetic reagent (*D. melanogaster*) | *EEC>* | https://doi.org/10.1038/s41467-022-28268-x | | *R57C10-GAL80-WPRE; Tub-GAL80^{ts}; voilà-GAL4* |
| Genetic reagent (*D. melanogaster*) | *moody-GAL4* | BDSC #90883 | RRID:BDSC_90883 | |
| Genetic reagent (*D. melanogaster*) | *R57C10-GAL4* | BDSC #39171 | RRID:BDSC_39171 | |
| Genetic reagent (*D. melanogaster*) | *R57C10-GAL80-WPRE* on X | Gift of Ryusuke Niwa | | |
| Genetic reagent (*D. melanogaster*) | *repo-GAL4* | BDSC #7415 | RRID:BDSC_7415 | |
| Genetic reagent (*D. melanogaster*) | *Tk::2A::GAL4* | BDSC #84693 | RRID:BDSC_84693 | |
| Genetic reagent (*D. melanogaster*) | *Tk^{Gut}>* | https://doi.org/10.1038/s42255-025-01267-0 | | *R57C10-GAL80-WPRE; Tk::2A::GAL4* |
| Genetic reagent (*D. melanogaster*) | *Tub-GAL80^{ts}* | BDSC #7108 | RRID:BDSC_7108 | |
| Genetic reagent (*D. melanogaster*) | *UAS-AstA-R1-RNAi^{KK}* | Vienna *Drosophila* Resource Center (VDRC) #101395 | | |
| Genetic reagent (*D. melanogaster*) | *UAS-AstA-R2-RNAi^{KK}* | VDRC #108648 | | |
| Genetic reagent (*D. melanogaster*) | *UAS-AstA-R2-RNAi^{TRiP}* | BDSC #67864 | RRID:BDSC_67864 | |
| Genetic reagent (*D. melanogaster*) | *UAS-AstA-RNAi^{KK}* | VDRC #103215 | | |
| Genetic reagent (*D. melanogaster*) | *UAS-AstA-RNAi^{TRiP}* | BDSC #25866 | RRID:BDSC_25866 | |
| Genetic reagent (*D. melanogaster*) | *UAS-dome-RNAi^{GD}* | VDRC #36356 | | |
| Genetic reagent (*D. melanogaster*) | *UAS-dome-RNAi^{KK}* | VDRC #106071 | | |
| Genetic reagent (*D. melanogaster*) | *UAS-dome-RNAi^{TRiP}* | BDSC #53890 | RRID:BDSC_53890 | |
| Genetic reagent (*D. melanogaster*) | *UAS-dsRed* | From BDSC #8546 | | |
| Genetic reagent (*D. melanogaster*) | *UAS-hop^{Tum}* | Gift of David Bilder | | |
| Genetic reagent (*D. melanogaster*) | *UAS-mCD8::GFP* | BDSC #5137 | RRID:BDSC_5137 | |
| Genetic reagent (*D. melanogaster*) | *UAS-TrpA1* | BDSC #26263 | RRID:BDSC_26263 | |
| Genetic reagent (*D. melanogaster*) | *UAS-upd2^{CRISPR} (attP2)* | This work | | |
| Genetic reagent (*D. melanogaster*) | *UAS-upd2-RNAi^{SH}* | VDRC #330691 | | |
| Genetic reagent (*D. melanogaster*) | *UAS-upd3* | Gift of David Bilder | | |

*Continued on next page*

*Continued*

| Reagent type (species) or resource | Designation | Source or reference | Identifiers | Additional information |
|---|---|---|---|---|
| Genetic reagent (*D. melanogaster*) | *UAS-upd3$^{CRISPR}$ (attP2)* | This work | | |
| Genetic reagent (*D. melanogaster*) | *UAS-upd3-RNAi$^{GD}$* | VDRC #27136 | | |
| Genetic reagent (*D. melanogaster*) | *UAS-upd3-RNAi$^{KK}$* | VDRC #106869 | | |
| Genetic reagent (*D. melanogaster*) | *UAS-upd3-RNAi$^{TRiP}$* | BDSC #32859 | RRID:BDSC_32859 | |
| Genetic reagent (*D. melanogaster*) | *upd2,3Δ* | Gift of Bruno Lemaitre | | |
| Genetic reagent (*D. melanogaster*) | *upd3-GAL4, UAS-GFP* | BDSC #98420 | RRID:BDSC_98420 | |
| Genetic reagent (*D. melanogaster*) | *upd3Δ* | Gift of Bruno Lemaitre | | |
| Genetic reagent (*D. melanogaster*) | *voilà-GAL4* | Gift of Alessandro Scopelliti | | |
| Genetic reagent (*D. melanogaster*) | *w$^{1118}$* | VDRC #60000 | | |
| Antibody | anti-AstA, rabbit polyclonal | Jena Bioscience, #ABD-062 | | IF(1:2000) |
| Antibody | anti-chicken, Alexa Fluor 488-conjugated goat polyclonal | Thermo Fisher, #A11039 | RRID:AB_2534096 | IF(1:500) |
| Antibody | anti-GFP, chicken polyclonal | Thermo Fisher, #A10262 | RRID:AB_2534023 | IF(1:500) |
| Antibody | anti-GFP, mouse monoclonal (3E6) | Thermo Fisher, #A11120 | RRID:AB_221568 | IF(1:500) |
| Antibody | anti-mCherry (used against Ds Red), rat monoclonal (16D7) | Thermo Fisher, #M11217 | RRID:AB_2536611 | IF(1:1000) |
| Antibody | anti-mouse, Alexa Fluor 488-conjugated goat polyclonal | Thermo Fisher, #A11001 | RRID:AB_2534069 | IF(1:500) |
| Antibody | anti-mouse, Alexa Fluor 555-conjugated goat polyclonal | Thermo Fisher, #A21422 | RRID:AB_2535844 | IF(1:500) |
| Antibody | anti-mouse, Alexa Fluor 647 Plus-conjugated goat polyclonal | Thermo Fisher, #A32728 | RRID:AB_2633277 | IF(1:500) |
| Antibody | anti-Prospero, mouse monoclonal | University of Iowa Developmental Studies Hybridoma Bank, #MR1A | Antibody Registry ID:AB_528440 | IF(1:20) |
| Antibody | anti-rat, Alexa Fluor 555-conjugated goat polyclonal | Thermo Fisher, #A21434 | RRID:AB_2535855 | IF(1:500) |
| Antibody | anti-Repo, mouse monoclonal | University of Iowa Developmental Studies Hybridoma Bank, #8D12 | Antibody Registry ID:AB_528448 | IF(1:50) |
| Recombinant DNA reagent | pCFD6 (UAS-CRISPR-gRNA plasmid) | Addgene #73915 | | |
| Sequence-based reagent | *AstA* forward oligo | This work | | Sequence provided in *Table 2* |
| Sequence-based reagent | *AstA* reverse oligo | This work | | Sequence provided in *Table 2* |
| Sequence-based reagent | *AstA-R1* forward oligo | This work | | Sequence provided in *Table 2* |
| Sequence-based reagent | *AstA-R1* reverse oligo | This work | | Sequence provided in *Table 2* |
| Sequence-based reagent | *AstA-R2* forward oligo | This work | | Sequence provided in *Table 2* |
| Sequence-based reagent | *AstA-R2* reverse oligo | This work | | Sequence provided in *Table 2* |
| Sequence-based reagent | *dome* forward oligo | This work | | Sequence provided in *Table 2* |
| Sequence-based reagent | *dome* reverse oligo | This work | | Sequence provided in *Table 2* |
| Sequence-based reagent | *Rp49* forward oligo | This work | | Sequence provided in *Table 2* |
| Sequence-based reagent | *Rp49* reverse oligo | This work | | Sequence provided in *Table 2* |
| Sequence-based reagent | *upd2* forward oligo | This work | | Sequence provided in *Table 2* |

*Continued on next page*

*Continued*

| Reagent type (species) or resource | Designation | Source or reference | Identifiers | Additional information |
|---|---|---|---|---|
| Sequence-based reagent | *upd2* reverse oligo | This work | | Sequence provided in *Table 2* |
| Sequence-based reagent | *upd3* forward oligo | This work | | Sequence provided in *Table 2* |
| Sequence-based reagent | *upd3* reverse oligo | This work | | Sequence provided in *Table 2* |
| Sequence-based reagent | *upd2* forward gRNA oligo for cloning of CRISPR construct | This work | | Sequence provided in *Table 1* |
| Sequence-based reagent | *upd2* reverse gRNA oligo for cloning of CRISPR construct | This work | | Sequence provided in *Table 1* |
| Sequence-based reagent | *upd3* forward gRNA oligo for cloning of CRISPR construct | This work | | Sequence provided in *Table 1* |
| Sequence-based reagent | *upd3* reverse gRNA oligo for cloning of CRISPR construct | This work | | Sequence provided in *Table 1* |
| Commercial assay or kit | Alexa Fluor 488-linked B3 hairpin | Molecular Instruments (Los Angeles, CA) | | |
| Commercial assay or kit | Alexa Fluor 546-linked B5 hairpin | Molecular Instruments (Los Angeles, CA) | | |
| Commercial assay or kit | B3-linked *upd3* probe set for hybridization chain reaction | Molecular Instruments (Los Angeles, CA) | | |
| Commercial assay or kit | B5-linked *prospero* probe set for hybridization chain reaction | Molecular Instruments (Los Angeles, CA) | | |
| Commercial assay or kit | cDNA synthesis kit | Applied Biosystems 'High-capacity cDNA synthesis kit', #4368814 | | |
| Commercial assay or kit | Gibson assembly kit | New England Biolabs 'NEBuilder HiFi DNA Assembly Master Mix', #E2621S | | |
| Commercial assay or kit | Q5 polymerase | New England Biolabs, #M0491S | | |
| Commercial assay or kit | qPCR master mix | Ampliqon 'Real Q Plus 2x Master Mix, Green', #A324402 | | |
| Commercial assay or kit | RNA extraction kit | Macherey-Nagel 'Nucleospin RNA kit', #740955 | | |
| Commercial assay or kit | Triacylglyceride measurement assay | Randox, #TR210 | | |
| Commercial assay or kit | TUNEL assay | Roche 'In Situ Cell Death Detection Kit, Fluorescein', #11684795910 | | |
| Chemical compound, drug | Erioglaucine dye | Sigma, #861146 | | |
| Chemical compound, drug | Fluoroshield mounting medium with DAPI | Nordic Biosite, #GTX30920 | | |
| Chemical compound, drug | ProLong Glass anti-fade mountant | Invitrogen, #36984 | | |
| Chemical compound, drug | SSC, 20x concentrate | Sigma, #S6639 | | |
| Chemical compound, drug | Triton X-100 | Merck, #12298 | | |
| Chemical compound, drug | Tween-20 | Sigma, #P1379 | | |
| Software, algorithm | ImageJ/FIJI image-analysis package, version 1.54p | Wayne Rasband and contributors, NIH, USA; http://imagej.org | | |
| Software, algorithm | Matlab coding environment, version R2023a Update 7 (9.14.0.2674353) | The MathWorks, Inc. | | |
| Software, algorithm | Matlab sleep-analysis script | https://doi.org/10.1371/journal.pgen.1008727 https://doi.org/10.1371/journal.pgen.1007623 | | |
| Software, algorithm | Prism statistics and presentation package, version 10.5.0 | GraphPad Software, LLC | | |
| Software, algorithm | Zen image-acquisition package, version Blue, 3.1 | Zeiss | | |
| Other | *Drosophila* Activity Monitoring System for sleep assays | TriKinetics (Waltham, MA) | | Infrared beam-break system for fly locomotor and sleep tracking. |
| Other | EnSight plate reader | PerkinElmer | | Plate reader for absorbance, fluorescence, and luminescence. |

*Continued on next page*

*Continued*

| Reagent type (species) or resource | Designation | Source or reference | Identifiers | Additional information |
|---|---|---|---|---|
| Other | Fly Liquid-food Interaction Counter (FLIC) apparatus for feeding assays | Sable Systems | | Automated fly feeding monitor via electrical contact detection. |
| Other | LSM-900 confocal microscope | Zeiss | | Confocal microscope for high-resolution fluorescence imaging. |
| Other | Poly-L-lysine coated glass microscope slides | Sigma, #P8920 | | Slides with poly-L-lysine for tissue/cell adhesion. |
| Other | QuantStudio 5 qPCR machine | Applied Biosystems | | Real-time PCR instrument. |
| Other | Stainless-steel mill balls | QIAGEN, #69989 | | Grinding tools for tissue disruption. |
| Other | TissueLyser LT bead mill | QIAGEN | | Bead mill homogenizer for tissue disruption. |

## *Drosophila* stocks and husbandry

Flies were cultured using a standard cornmeal-based formulation (82 g/l cornmeal, 60 g/l sucrose, 34 g/l yeast, 8 g/l agar, 4.8 ml/l propionic acid, and 1.6 g/l methyl-4-hydroxybenzoate) maintained at 25°C with 60% relative humidity under a 12-hr light/dark cycle. Post-eclosion, flies were transitioned to an adult-specific, cornmeal-free diet (comprising 90 g/l sucrose, 80 g/l yeast, 10 g/l agar, 5 ml/l propionic acid, and 15 ml/l of a 10% methyl-4-hydroxybenzoate solution in ethanol) (*Tennessen et al., 2014*) for 4–7 days prior to experiments. Adult mated females were used for all experiments. Flies were separated by sex 1 day prior to experimental procedures. Strains harboring the temperature-sensitive *Tubulin-GAL80^ts* transgene were initially reared at 18°C on cornmeal food and then switched to the adult diet for 3–4 days post-eclosion, still at 18°C. Subsequently, they were incubated at 29°C for 5–7 days to activate RNAi expression in advance of the experiments. To ensure optimal conditions, the flies were provided with fresh food every 2-3 days. The following lines used in this study were sourced from the Bloomington *Drosophila* Stock Center (BDSC) at the University of Indiana: *R57C10-GAL4* (#39171); *UAS-upd3-RNAi^TRiP* (#32859); *UAS-dome-RNAi^TRiP* (#53890); *AstA::2A::GAL4* (#84593); *AstA-R1::2A::GAL4* (#84709); *AstA-R2::2A::GAL4* (#84594); *UAS-AstA-RNAi^TRiP* (#25866); *UAS-AstA-R2-RNAi^TRiP* (#67864); *UAS-mCD8::GFP* (#5137); *Tub-GAL80^ts* (#7108); *repo-GAL4* (#7415); *moody-GAL4* (#90883); *UAS-TrpA1* (#26263); *UAS-dsRed* was extracted from (#8546); *AstC::2A::GAL4* (#84595); *Tk::2A::GAL4* (#84693); and *upd3-GAL4, UAS-GFP* (#98420; GFP variant and protein localization are unknown in this line). Additional fly lines were acquired from the Vienna *Drosophila* Resource Center (VDRC): control line *w^1118* (#60000, which is isogenic with the VDRC RNAi lines); *UAS-upd2-RNAi^SH* (#330691); *UAS-upd3-RNAi^KK* (#106869); *UAS-upd3-RNAi^GD* (#27136); *UAS-dome-RNAi^KK* (#106071); *UAS-dome-RNAi^GD* (#36356); *UAS-AstA-RNAi^KK* (#103215); *UAS-AstA-R1-RNAi^KK* (#101395); *UAS-AstA-R2-RNAi^KK* (#108648). The *upd3Δ* and *upd2,3Δ* deletion mutants were kindly provided by Bruno Lemaitre. *UAS-upd3* and *UAS-hop^Tum* lines were gifts from David Bilder. The *6xSTAT-dGFP:2A::RFP* line was generously supplied by Norbert Perrimon. The *voilà-GAL4* strain was graciously provided by Alessandro Scopelliti. The *R57C10-GAL80* transgene, situated on the X chromosome, was kindly donated by Ryusuke Niwa. The *10xSTAT-GFP* line was a gift from Julien Colombani. To ensure uniformity in genetic background and to create control groups with an appropriate genetic background, all *GAL4* and *GAL80* lines used in this study were backcrossed to a *w^1118* line for multiple generations before being outcrossed with the genetic background specific to the RNAi, CRISPR, or overexpression lines to serve as controls in the experiments (*Kubrak et al., 2022*). This ensures that the only difference between experimental and control animals is the presence or absence of the UAS transgene, providing the most appropriate control for assessing transgene-specific effects.

## Generation of tissue-specific CRISPR lines

To facilitate tissue-specific CRISPR-based disruption of the *upd2* and *upd3* loci, constructs were prepared containing two gRNA target sequences, flanked by efficiency-enhancing tRNA sequences. One construct was prepared for *upd2*, and two transgenes, targeting different genomic sites, were made for *upd3*. The *upd2* construct was designed to delete the region encoding the secreted Upd2 protein. One *upd3* construct should delete the initiator ATG codon, and the other – the one used in this work – deletes the second exon, which contains a significant portion of the coding sequence.

**Table 1.** Oligos used for cloning the upd2 and upd3 CRISPR constructs, with gRNA sequences indicated in bold and underlined.

| Oligo | Sequence |
|---|---|
| *upd2fwd* | CGGCCCGGGTTCGATTCCCGGCCGATGCA**TTCTCGCCCGCTCGATTGGT**GTTTCAGAGCTATGCTGGAAAC |
| *upd2rev* | ATTTTAACTTGCTATTTCTAGCTCTAAAAC**CATGCAACAGTCACTGACGA**TGCACCAGCCGGGAATCGAACC |
| *upd3fwd* | CGGCCCGGGTTCGATTCCCGGCCGATGCA**GACAACTGAACTGAACCGAC**GTTTCAGAGCTATGCTGGAAAC |
| *upd3rev* | ATTTTAACTTGCTATTTCTAGCTCTAAAAC**TTTGGTTCTGTAGATTCTGC**TGCACCAGCCGGGAATCGAACC |

Target-sequence cassettes were assembled by first cloning the tRNA insert from plasmid pCFD6 (Addgene #73915) between long oligos containing the gRNA target sequences using Q5 polymerase (New England Biolabs, #M0491S). The vector and the PCR products were then integrated using Gibson assembly (NEBuilder HiFi DNA Assembly Master Mix, New England Biolabs, #E2621S). Clones were sequenced to verify accuracy, and correct constructs were integrated into the fly genome at the attP2 site (chromosome 3L) by BestGene (Chino Hills, CA). The sequences used for cloning the *upd2* and *upd3* CRISPR constructs, with gRNA sequences indicated in bold, are shown in *Table 1*.

## Sleep, activity, and survival assays

The *Drosophila* Activity Monitoring System (TriKinetics, Waltham, MA) was employed to track sleep and activity patterns. Single flies aged 6–8 days after eclosion were placed into glass tubes using light $CO_2$ anesthesia. On one end, the tubes were sealed with a foam plug; on the other was placed a detachable 250 µl PCR tube containing 90 µl of feeding medium: either 5% sucrose in 1% agar/water, 5% sucrose mixed with various concentrations of $H_2O_2$ in 1% agar/water, or plain 1% agar/water for starvation conditions. All food media contained 0.5% propionic acid and 0.15% methyl-4-hydroxybenzoate to prevent microbial growth, with $H_2O_2$ being supplemented once the food had cooled to below 40°C. Monitoring of the flies' locomotor activity and sleep began at the beginning of the light cycle, after the animals had spent their first day in the tubes acclimating. Following an additional 24 hr on the standard 5%-sucrose diet, the PCR tubes were replaced with fresh ones containing $H_2O_2$ or starvation media at the lights-on transition when animals were awake, to avoid unnecessary disturbances to the animals. For recovery experiments, animals were switched back to a 5%-sucrose diet after 24 hr on $H_2O_2$-laced food. Periods of inactivity lasting 5 min or longer were recorded as 'sleep'. In the sleep deprivation studies, the flies were placed in DAM monitors and subjected to mechanical stimulation, which was produced by attaching the monitors to a vortexer mounting plate (TriKinetics) and vibrating them for 2 s at the start of each minute throughout the 6-hr interval leading up to the lights-on time. Recovery sleep was assessed in flies that experienced a reduction of more than 60% in their typical sleep during the deprivation period, using their sleep patterns from the 24-hr period before the onset of sleep deprivation as a baseline. The occurrence of recovery sleep was specifically evaluated during the first 2 hr immediately following the sleep deprivation phase. For survival assays, flies were loaded into tubes filled with either plain 1% agar/water for starvation or 1% $H_2O_2$ in 1% agar/water to test oxidative stress resistance. The time of death was recorded upon the complete cessation of movement.

## Feeding assays

Short-term food consumption was quantified using a spectrophotometric dye-feeding assay (*Wong et al., 2009*; *Skorupa et al., 2008*). All food intake experiments were conducted during the time of the normal morning meal, 1 hr after lights-on in a 12:12 hr dark/light cycle. Flies were transferred without anesthesia to food (90 g/l sucrose, 80 g/l yeast, 10 g/l agar, 5 ml/l propionic acid, and 15 ml/l of a 10% methyl-4-hydroxybenzoate solution in ethanol) supplemented with 0.5% erioglaucine dye (brilliant blue R, FD&C Blue No. 1, Sigma-Aldrich, #861146) and allowed to feed for 1 hr. A control group of flies was provided with undyed food to establish the baseline absorbance levels of fly lysates. For each genotype, 1–2 flies per sample were homogenized in 100 µl phosphate buffer (pH 7.5) using a TissueLyser LT (QIAGEN) with 5 mm stainless-steel beads. Homogenates were centrifuged at 16,000 × *g* for 5 min, and 50 µl of the cleared supernatant was transferred to a 384-well plate. Absorbance was measured at 629 nm for erioglaucine using an Ensight multi-mode plate reader (PerkinElmer).

Standard curves for dye were employed to correlate absorbance readings with the amounts of food consumed.

To assess feeding behavior, interactions with food were monitored over a 20- to 24-hr period using the Fly Liquid-Food Interaction Counter (FLIC) apparatus (*Ro et al., 2014*). *Drosophila* Feeding Monitors (DFMs; Sable Systems) were placed in an incubator set to 25°C (or 29°C for strains carrying *GAL80^{ts}*), maintaining 70% humidity under a 12:12-hr light/dark cycle. Each of the 12 DFM chambers was filled with a 10% sucrose solution, and individual flies were introduced in the afternoon following the morning meal. After several hours of acclimation, evening feeding activity was recorded. The following morning, at lights-on, the DFMs were refilled with fresh sugar solution, and data from the morning meal were collected. The feeding behavior was recorded using the manufacturer's software and analyzed using R Studio with the provided package (https://github.com/PletcherLab/FLIC_R_Code, *Pletcher, 2024*).

## Immunohistochemistry, TUNEL staining, and confocal imaging

Adult midguts, brains, and VNCs were dissected in cold PBS and fixed for 1 hr at room temperature in 4% paraformaldehyde/PBS with gentle shaking. After a quick rinse with PBST (PBS with 0.1% Triton X-100, Merck #12298), the tissues were washed three times for 15 min each in PBST. For TUNEL staining, the In Situ Cell Death Detection Kit, Fluorescein (Roche, #11684795910), was used according to the manufacturer's instructions, and tissues were subsequently washed in PBS before mounting. For samples undergoing antibody staining, tissues were then blocked for 30 min at room temperature in PBST containing 5% normal goat serum (Sigma) and subsequently incubated overnight (or 2 days for CNS samples) at 4°C with primary antibodies diluted in the blocking solution with gentle agitation. After removing the primary antibody solution, tissues were rinsed once and washed three times for 20 min each in PBST. Secondary antibodies diluted in PBST were applied, and tissues were incubated overnight at 4°C, followed by three PBST washes and one PBS wash. The samples were then mounted on poly-L-lysine-coated slides (Sigma, #P8920) in Fluoroshield mounting medium with DAPI (Nordic Biosite, #GTX30920), and imaged on a Zeiss LSM-900 confocal microscope using a 20× air or 40× oil objective with Zen software. Image stitching was performed using the Stitching function of Zeiss Zen Blue 3.1, and analysis was conducted using the open-source FIJI/ImageJ software package (*Schindelin et al., 2012*). All samples compared with each other within a figure panel were dissected, stained, and imaged simultaneously using identical settings and reagents. For quantification of AstA peptide levels in the R5 region of the posterior midgut, anti-AstA stained images were processed in FIJI, and mean fluorescence intensity was measured in the R5 region. Background signal was subtracted using neighboring non-AstA-expressing areas from the same tissue. To quantify 6xSTAT-dGFP::2A::RFP expression (temporally resolved JAK/STAT activity indicator), dissected brains were fixed for 10 min in 4% paraformaldehyde at room temperature with agitation, rinsed once with PBST, mounted, and imaged immediately without antibody staining. For 10xSTAT-GFP quantification, brain and VNC samples were stained as described above, Z-stacks were projected in FIJI using the 'sum' method, and Repo-positive cells at the surface were manually segmented to measure the raw integrated density with local background subtraction. For measuring 10xSTAT-GFP across the BBB, linear regions of interest were drawn through the glial layer perpendicular to the brain surface at the plane showing maximum brain size using FIJI's line tool; GFP intensity was then quantified along these lines, and the peak was recorded for each transect. Antibodies used included rabbit anti-AstA (Jena Bioscience, #ABD-062, no RRID; diluted 1:2000), mouse anti-Repo (University of Iowa Developmental Studies Hybridoma Bank, #8D12, no RRID, Antibody Registry ID AB_528448; diluted 1:50), mouse anti-Prospero (University of Iowa Developmental Studies Hybridoma Bank, #MR1A, no RRID, Antibody Registry #AB_528440; diluted 1:20), mouse anti-GFP (clone 3E6, Thermo Fisher, #A11120, RRIDAB_221568; diluted 1:500), chicken anti-GFP (Thermo Fisher, #A10262, RRIDAB_2534023; diluted 1:500), rat anti-mCherry (used against dsRed; clone 16D7, Thermo Fisher, #M11217, RRIDAB_2536611; diluted 1:1000), Alexa Fluor 488-conjugated goat anti-mouse (Thermo Fisher, #A11001, RRIDAB_2534069; diluted 1:500), Alexa Fluor 555-conjugated goat anti-mouse (Thermo Fisher, #A21422, RRIDAB_2535844; diluted 1:500), Alexa Fluor 647 Plus-conjugated goat anti-mouse (Thermo Fisher, #A32728, RRIDAB_2633277; diluted 1:500), Alexa Fluor 555-conjugated goat anti-rat (Thermo Fisher, #A21434, RRIDAB_2535855; diluted 1:500), and 488-conjugated goat anti-chicken (Thermo Fisher, #A11039, RRIDAB_2534096; diluted 1:500).

## In situ hybridization

To detect co-expression of specific mRNAs, we performed fluorescent in situ hybridization using the hybridization chain reaction (HCR) method. This approach, based on a previously published protocol (*Bruce et al., 2012*) with slight modifications, utilizes fluorescent probes and reagents from Molecular Instruments (Los Angeles, CA, USA). Adult *Drosophila* midguts were dissected in ice-cold PBS and fixed in 4% paraformaldehyde for 1 hr at room temperature with gentle rocking. After fixation, tissues were rinsed and washed three times for 10 min each at room temperature in PBS containing 0.1% Tween-20 (Sigma, #P1379). To permeabilize the tissue, samples were incubated at 37°C for 30 min in a buffer containing 1% SDS, 0.5% Tween-20, 50 mM Tris-HCl, 1 mM EDTA, and 150 mM NaCl at pH 7.5. Tissues were then incubated in pre-warmed hybridization buffer (from the HCR buffer kit) for 30 min at 37°C. Hybridization was carried out overnight at the same temperature using hybridization buffer containing 15 nM each of B3-labeled *upd3* probes and B5-labeled *prospero* probes (Molecular Instruments). The next day, tissues were washed four times for 15 min each with pre-warmed probe wash buffer, followed by two 5 min washes in 5×SSCT (prepared using 20× SSC concentrate [Sigma, #S6639] with 0.1% Tween-20 [Sigma, #P1379]). Amplification was initiated by incubating tissues in amplification buffer for 40 min at room temperature, followed by an overnight incubation with 120 nM of each corresponding fluorophore-labeled hairpin (Alexa Fluor 488-B3 and Alexa Fluor 546-B5), in the dark at room temperature. After amplification, tissues were washed five times for 10 min each in 5× SSCT, followed by six 15 min washes in PBS. Midguts were mounted on poly-L-lysine–coated slides (Sigma, #P8920) using ProLong Glass Antifade Mountant (Invitrogen, #36984), and coverslipped using a 0.12-mm spacer and 0.1-mm glass coverslip. Imaging was performed with a Zeiss LSM-900 confocal microscope using Zen software.

## TAG measurements

TAG concentrations were determined following established methods (*Tennessen et al., 2014*; *Hildebrandt et al., 2011*; *Kubrak et al., 2024*) using the Randox Triglycerides (GPO-PAP) method (Randox, #TR210). For each sample, flies were homogenized in 50 µl PBS per fly (between 2 and 4 flies per sample) containing 0.1% Tween-20 (Sigma #1379) using a TissueLyser LT (QIAGEN) with 5 mm stainless-steel beads, 50 oscillations/s for 30 s. Homogenates were heated at 70°C for 10 min to inactivate endogenous enzymes and centrifuged at 11,000 × *g* for 1 min. Aliquots of cleared and vortexed supernatants (4 µl) were added to 36 µl of triglyceride reagent (Randox, #TR210) in a 384-well plate, covered with ultra-high-clarity optical film (ThermalSeal RT2RR, Z722553, Excel Scientific). The plate was spun down at 1500 × g for 1 min to settle fluids and eliminate bubbles and incubated for 10 min at room temperature. Absorbance for each sample was measured at 540 nm on an Ensight multimode

**Table 2.** List of qPCR primers.

| Gene | Primer | Sequence |
|------|--------|----------|
| *upd2* | Forward | CGGAACATCACGATGAGCGAA |
| *upd2* | Reverse | TCGGCAGGAACTTGTACTCG |
| *upd3* | Forward | TGTCGAGAAGAACAAGTGGCG |
| *upd3* | Reverse | CGTGGCGAAGGTTCAACTGT |
| *dome* | Forward | CTCACGTCTCGACTGGGAAC |
| *dome* | Reverse | AGAATGGTGCTTGTCAGGCA |
| *AstA* | Forward | CGCCTGCCGGTCTATAACTT |
| *AstA* | Reverse | CTTGTTCTGTCGGCCAGGTC |
| *AstA-R1* | Forward | GCCACTGGAAACGGTAGTATC |
| *AstA-R1* | Reverse | CGTGTGTTCCGAGGTGAATG |
| *AstA-R2* | Forward | CGCAGTGTCCAGTACCTGATT |
| *AstA-R2* | Reverse | GAGCGAATGGGATGAACCAC |
| *Rp49* | Forward | AGTATCTGATGCCCAACATCG |
| *Rp49* | Reverse | CAATCTCCTTGCGCTTCTTG |

plate reader (PerkinElmer). The readings were then converted to TAG concentrations using standard curves, prepared with triglyceride standards (Randox, 1352TR CAL Standard).

## Measurement of transcript levels using qPCR

Several tissue samples containing three dissected guts, brains, or heads were collected for each condition or genotype. These samples were then homogenized in 2 ml Eppendorf tubes filled with lysis buffer containing 1% beta-mercaptoethanol, utilizing a TissueLyser LT bead mill (QIAGEN) with 5 mm stainless steel beads (QIAGEN #69989). RNA extraction was carried out with the NucleoSpin RNA kit (Macherey-Nagel, #740955) following the guidelines provided by the manufacturer. cDNA was synthesized using the High-Capacity cDNA Synthesis kit (Applied Biosystems, #4368814). Quantitative PCR was performed with RealQ Plus 2x Master Mix Green (Ampliqon, #A324402) using a QuantStudio 5 (Applied Biosystems) instrument. Gene expression results were normalized to the housekeeping gene *Rp49* using the delta-delta-Ct method. The specific oligonucleotides used are given in *Table 2*.

## Statistics

Statistical analyses were performed using the Prism software package (GraphPad, version 10). Data were tested for normality before assessments of significance. For data following a normal distribution, pairwise analyses were conducted using two-tailed unpaired Student's *t*-tests, and comparisons involving multiple samples used one-way ANOVA with subsequent post hoc tests for multiple comparisons. Non-normally distributed data were analyzed using two-tailed unpaired Mann–Whitney *U* tests or one-way Kruskal–Wallis ANOVA, followed by multiple comparisons. Additionally, interactions between genotype and diet were calculated using two-way ANOVA. All plots represent the mean ± SEM. All replicates represent independent biological samples.

## Code availability

The MatLAB scripts used for analyzing sleep are described in *Maurer et al., 2020*; *Nagy et al., 2018*.

## Acknowledgements

This work was supported by Lundbeck Foundation grant 2019-772 and Novo Nordisk Foundation grant NNF19OC0054632 to KR. The Zeiss LSM 900 confocal microscope and the PerkinElmer EnSight plate reader were purchased with generous grants from the Carlsberg Foundation (Nos. CF19-0353 and CF17-0615, respectively) to KR.

---

## Additional information

### Funding

| Funder | Grant reference number | Author |
|---|---|---|
| Lundbeck Foundation | 2019-772 | Kim Rewitz |
| Novo Nordisk Fonden | NNF19OC0054632 | Kim Rewitz |
| Carlsbergfondet | CF19-0353 | Kim Rewitz |
| Carlsbergfondet | CF17-0615 | Kim Rewitz |

The funders had no role in study design, data collection, and interpretation, or the decision to submit the work for publication.

### Author contributions

Alina Malita, Conceptualization, Data curation, Formal analysis, Supervision, Validation, Investigation, Visualization, Methodology, Writing – original draft, Project administration, Writing – review and editing; Anne H Skakkebaek, Olga Kubrak, Xiaokang Chen, Nadja Ahrentloev, Formal analysis, Investigation; Takashi Koyama, Methodology; Elizabeth C Connolly, Formal analysis, Investigation, Methodology; Ditte S Andersen, Provided input and contributed to editing the manuscript; Michael J Texada, Writing – review and editing, Provided input and contributed to editing the manuscript; Kenneth

Halberg, Provided input and contributed to editing the manuscript; Kim Rewitz, Conceptualization, Supervision, Funding acquisition, Writing – original draft, Project administration, Writing – review and editing

### Author ORCIDs
Alina Malita https://orcid.org/0000-0003-1333-1944
Anne H Skakkebaek https://orcid.org/0009-0004-6793-4809
Olga Kubrak https://orcid.org/0009-0008-2540-6628
Takashi Koyama https://orcid.org/0000-0003-4203-114X
Elizabeth C Connolly https://orcid.org/0000-0002-5716-8889
Ditte S Andersen https://orcid.org/0000-0003-1209-6102
Michael J Texada https://orcid.org/0000-0003-2479-1241
Kenneth Halberg https://orcid.org/0000-0002-5903-7196
Kim Rewitz https://orcid.org/0000-0002-4409-9941

Joint Public Review: https://doi.org/10.7554/eLife.99999.3.sa1
Author response https://doi.org/10.7554/eLife.99999.3.sa2

---

## Additional files

### Supplementary files
MDAR checklist

Source data 1. Source data for all figures in the manuscript, including raw measurements and processed values used for plots and statistical analyses.

### Data availability
All data supporting the findings of this study are provided within the article and its supplementary information files. A source data file has been provided for all figures.

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
