## [Editor Report · eLife Assessment]

This **important** work by Malita et al. describes a mechanism by which an intestinal inflammatory response causes an increase in daytime sleep through signaling from the gut to the blood-brain barrier. Their findings suggest that cytokines Upd3 and Upd2 produced by the intestine following inflammation act on glia of the blood brain barrier to regulate sleep by modulating Allatostatin A signaling. The evidence is **compelling** and elegantly performed using the ample *Drosophila* genetic toolbox, making this work appealing for a broad group of neuroscience researchers interested in sleep and gut-brain interactions.

---

## [Referee Report · Joint Public Review]

Summary:

Malita and colleagues investigated the mechanism by which infections increase sleep in Drosophila. Their work is important because it further supports the idea that the blood brain barrier is involved in brain-body communication, and because it advances the field of sleep research. Using knock-down and knock-out of cytokines and cytokine receptors specifically in the endocrine cells of the gut (cytokines) as well as in the glia forming the blood-brain barrier (BBB) (cytokines receptors), the authors show that cytokines, upd2 and upd3, secreted by entero-endocrine cells in response to infections increase sleep through the Dome receptor in the BBB. They also show that gut-derived Allatostatin (Alst) A promotes wakefulness by inhibiting the Alst A signaling that is mediated by Alst receptors expressed in BBB glia. Their results suggest there may be additional mechanisms that promote elevated sleep during gut inflammation. The evidence supporting most of their claims is compelling. Nevertheless, the activation of the sleep-promoting pathway by infection should be accomplished through bacterial infection of the gut.

Strengths:

The work is, in general, supported by well-designed and well-performed experiments, especially those that show that the endocrine cells from the gut are the sources of the Upd cytokines, the effects of these cytokines on daytime sleep, and that the glial cells of the BBB are the target cell for the Upds action. In addition, the evidence associating the downregulation of Alst receptors in the BBB by Upd and Jak/Stat pathways is compelling.

Weaknesses:

(1) The model of gut inflammation that is used is based on the increase in reactive oxygen species (ROS) that is caused by adding 1% H2O2 to the food. The use of the model is supported rather weakly by two papers (ref. 26 and 27). The paper by Jiang et al. (26) shows that the infection by Pseudomonas entomophila induces cytokine responses Upd2 and 3, which are also induced by the Jnk pathway; there is no mention of ROS. Buchon et al. (27) is a review that refers to results that indicate that as part of the immune response to pathogens in the gut, there is production of ROS by the NADPH oxidase DUOX. Thus, there is no strong support for the use of this model.

(2) There is no support for the use of ROS in the food instead a direct infection by pathogenic bacteria. It is known that ROS causes damage in the gut epithelium, which in turn induces the expression of the cytokines studied, which might be independent of infection and confound the results.

---

## [Author Response]

The following is the authors’ response to the original reviews.

**Joint Public Review:**
Summary:The authors sought to elucidate the mechanism by which infections increase sleep in Drosophila. Their work is important because it further supports the idea that the blood-brain barrier is involved in brain-body communication, and because it advances the field of sleep research. Using knock-down and knock-out of cytokines and cytokine receptors specifically in the endocrine cells of the gut (cytokines) as well as in the glia forming the blood-brain barrier (BBB) (cytokines receptors), the authors show that cytokines, upd2 and *upd3*, secreted by entero-endocrine cells in response to infections increase sleep through the Dome receptor in the BBB. They also show that gut-derived Allatostatin (Alst) A promotes wakefulness by inhibiting Alst A signaling that is mediated by Alst receptors expressed in BBB glia. Their results suggest there may be additional mechanisms that promote elevated sleep during gut inflammation.The authors suggest that *upd3* is more critical than upd2, which is not sufficiently addressed or explained. In addition, the study uses the gut's response to reactive oxygen molecules as a proxy for infection, which is not sufficiently justified. Finally, further verification of some fundamental tools used in this paper would further solidify these findings making them more convincing.Strengths:(1) The work addresses an important topic and proposes an intriguing mechanism that involves several interconnected tissues. The authors place their research in the appropriate context and reference related work, such as literature about sickness-induced sleep, ROS, the effect of nutritional deprivation on sleep, sleep deprivation and sleep rebound, upregulated receptor expression as a compensatory mechanism in response to low levels of a ligand, and information about Alst A.(2) The work is, in general, supported by well-performed experiments that use a variety of different tools, including multiple RNAi lines, CRISPR, and mutants, to dissect both signal-sending and receiving sides of the signaling pathway.(3) The authors provide compelling evidence that shows that endocrine cells from the gut are the source of the upd cytokines that increase daytime sleep, that the glial cells of the BBB are the targets of these upds, and that upd action causes the downregulation of Alst receptors in the BBB via the Jak/Stat pathways.

We are pleased that the reviewers recognized the strength and significance of our findings describing a gut-to-brain cytokine signaling mechanism involving the blood-brain barrier (BBB) and its role in regulating sleep, and we thank them for their comments.

Weaknesses:(1) There is a limited characterization of cell types in the midgut which are classically associated with upd cytokine production.

We thank the reviewer for raising this point. Although several midgut cell types (including the absorptive enterocytes) may indeed produce Unpaired (Upd) cytokines, our study specifically focused on enteroendocrine cells (EECs), which are well-characterized as secretory endocrine cells capable of exerting systemic effects. As detailed in our response to Results point #2 (please see below), we show that EEC-specific manipulation of Upd signaling is both necessary and sufficient to regulate sleep in response to intestinal oxidative stress. These findings support the role of EECs as a primary source of gut-derived cytokine signaling to the brain. To acknowledge the possible involvement of other source, we have also added a statement to the Discussion in the revised manuscript noting that other, non-endocrine gut cell types may contribute to systemic Unpaired signaling that modulates sleep.

(2) Some of the main tools used in this manuscript to manipulate the gut while not influencing the brain (e.g., Voilà and Voilà + R57C10-GAL80), are not directly shown to not affect gene expression in the brain. This is critical for a manuscript delving into intra-organ communication, as even limited expression in the brain may lead to wrong conclusions.

We agree with the reviewer that this is an important point. To address it, we performed additional validation experiments to assess whether the *voilà-GAL4* driver in combination with *R57C10-GAL80 (EEC>)* influences *upd2* or *upd3* expression in the brain. Our results show that manipulation using *EEC>* alters *upd2* and *upd3* expression in the gut (Fig. 1a,b), with new data showing that this does not affect their expression levels in neuronal tissues (Fig. S1a), supporting the specificity of our approach. These new data are now included in the revised manuscript and described in the Results section. This additional validation strengthens our conclusion that the observed sleep phenotypes result from gut-specific cytokine signaling, rather than from effects on Unpaired cytokines produced in the brain.

(1) >(3) The model of gut inflammation used by the authors is based on the increase in reactive oxygen species (ROS) obtained by feeding flies food containing 1% H2O2. The use of this model is supported by the authors rather weakly in two papers (refs. 26 and 27): The paper by Jiang et al. (ref. 26) shows that the infection by Pseudomonas entomophila induces cytokine responses upd2 and 3, which are also induced by the Jnk pathway. In addition, no mention of ROS could be found in Buchon et al. (ref 27); this is a review that refers to results showing that ROS are produced by the NADPH oxidase DUOX as part of the immune response to pathogens in the gut. Thus, there is no strong support for the use of this model.

We thank the reviewer for raising this point. We agree that the references originally cited did not sufficiently justify the use of H_2_O_2_ feeding as a model of gut inflammation. To address this, we have revised the Results section to clarify that we use H_2_O_2_ feeding as a controlled method to elevate intestinal ROS levels, rather than as a general model of inflammation. This approach allows us to investigate the specific effects of ROS-induced cytokine signaling in the gut. We have also added additional citations to support the physiological relevance of this model. For instance, Tamamouna et al. (2021) demonstrated that H_2_O_2_ feeding induces intestinal stem-cell proliferation – a response also observed during bacterial infection – and Jiang et al. (2009) showed that enteric infections increase *upd2* and *upd3* expression, which we similarly observe following H_2_O_2_ feeding (Fig. 3a). These findings support the use of H_2_O_2_ as a tool to mimic specific ROS-linked responses in the gut. We believe this targeted and tractable model is a strength of our study, enabling us to dissect how intestinal ROS modulates systemic physiology through cytokine signaling

Additionally, we have included a statement in the Discussion acknowledging that ROS generated during infection may activate signaling mechanisms distinct from those triggered by chemically induced oxidative stress, and that exploring these differences in future studies may yield important insights into gut–brain communication. These revisions provide a stronger justification for our model while more accurately conveying both its relevance and its limitations.

(2) >(4) Likewise, there is no support for the use of ROS in the food instead a direct infection by pathogenic bacteria. Furthermore, it is known that ROS damages the gut epithelium, which in turn induces the expression of the cytokines studied. Thus the effects observed may not reflect the response to infection. In addition, Majcin Dorcikova et al. (2023). Circadian clock disruption promotes the degeneration of dopaminergic neurons in male Drosophila. Nat Commun. 2023 14(1):5908. doi: 10.1038/s41467-02341540-y report that the feeding of adult flies with H2O2 results in neurodegeneration if associated with circadian clock defects. Thus, it would be important to discuss or present controls that show that the feeding of H2O2 does not cause neuronal damage.

We thank the reviewer for this thoughtful follow-up point. We would like to clarify that we do not claim that the effects observed in our study directly reflect the full response to enteric infection. As outlined in our revised response to comment 3, we have updated the manuscript to more precisely describe the H_2_O_2_-feeding paradigm as a model that induces local intestinal ROS responses comparable to, but not equivalent to, those observed during pathogenic challenges. This revised framing highlights both the potential similarities and differences between chemically induced oxidative stress and infection-induced responses. Indeed, in the revised Discussion, we now explicitly acknowledge that ROS generated during infection may engage distinct signaling mechanisms compared to exogenous H_2_O_2_ and emphasize the value of future studies in delineating these pathways. We are currently pursuing this direction in an independent ongoing study investigating the effects of enteric infections. However, for the present work, we chose to focus on the effects of ROS-induced responses in isolation, as this provides a clean and well-controlled context to dissect the specific contribution of oxidative stress to cytokine signaling and sleep regulation.

To further address the reviewer’s concern, we have also included new data (a TUNEL stain for apoptotic DNA fragmentation) in the revised manuscript showing that H_2_O_2_ feeding does not damage neuronal tissues under our experimental conditions (Fig. S3f,g). This addresses the point raised regarding the potential neurotoxicity of H_2_O_2_, as described by Majcin Dorcikova et al. (2023), and supports the specificity of the sleep phenotypes observed in our study. We believe these revisions and clarifications strengthen the manuscript and make our interpretation more precise.

(3) >(5) The novelty of the work is difficult to evaluate because of the numerous publications on sleep in Drosophila. Thus, it would be very helpful to read from the authors how this work is different and novel from other closely related works such as: Li et al. (2023) Gut AstA mediates sleep deprivation-induced energy wasting in Drosophila. Cell Discov. 23;9(1):49. doi: 10.1038/s41421-023-00541-3.

Our work highlights a distinct role for gut-derived AstA in sleep regulation compared to findings by Lin et al. (Cell Discovery, 2023)[1], who showed that gut AstA mediates energy wasting during sleep deprivation. Their study focused on the metabolic consequences of sleep loss, proposing that sleep deprivation increases ROS in the gut, which then promotes the release of the glucagon-like hormone adipokinetic hormone (AKH) through gut AstA signaling, thereby triggering energy expenditure.

In contrast, our study addresses the inverse question – how ROS in the gut influences sleep. In our model, intestinal ROS promotes sleep, raising the intriguing possibility – cleverly pointed out by the reviewers – that ROS generated during sleep deprivation might promote sleep by inducing Unpaired cytokine signaling in the gut. According to our findings, this suppresses wake-promoting AstA signaling in the BBB, providing a mechanism to promote sleep as a restorative response to gut-derived oxidative stress and potentially limiting further ROS accumulation. Importantly, our findings support a wakepromoting role for EEC-derived AstA, demonstrated by several lines of evidence. First, EEC-specific knockdown of *AstA* increases sleep. Second, activation of AstA^+^ EECs using the heat-sensitive cation channel Transient Receptor Potential A1 (TrpA1) reduces sleep, and this effect is abolished by simultaneous knockdown of *AstA*, indicating that the sleep-suppressing effect is mediated by AstA and not by other peptides or secreted factors released by these cells. Third, downregulation of *AstA* receptor expression in BBB glial cells increases sleep, further supporting the existence of a functional gut AstA– glia arousal pathway. We have now included new data in the revised manuscript showing that AstA release from EECs is downregulated during intestinal oxidative stress (Fig. 7k,l,m). This suggests that this wake-promoting signal is suppressed both at its source (the gut endocrine cells), by unknown means, and at its target, the BBB, via Unpaired cytokine signaling that downregulates AstA receptor expression. This coordinated downregulation may serve to efficiently silence this arousal-promoting pathway and facilitate sleep during intestinal stress. These new data, along with an expanded discussion, provide further mechanistic insight into gut-derived AstA signaling and strengthen our proposed model.

This contrasts with the interpretation by Lin et al., who observed increased AstA peptide levels in EECs after antioxidant treatment and interpreted this as peptide retention. However, peptide accumulation may result from either increased production or decreased release, and peptide levels alone are insufficient to distinguish between these possibilities. To resolve this, we examined *AstA* transcript levels, which can serve as a proxy for production. Following oxidative stress (24 h of 1% H_2_O_2_ feeding and the following day), when animals show increased sleep (Fig. 7e), we observed a decrease in *AstA* transcript levels followed by an increase in peptide levels (Fig. 7k,l,m), suggesting that oxidative stress leads to reduced gut AstA production and release. Furthermore, we recently found that a class of EECs that produce the hormone Tachykinin (Tk) and are distinct from the AstA^+^ EECs express the ROSsensitive cation channel *TrpA1* (Ahrentløv et al., 2025, Nature Metabolism2). In these Tk^+^ EECs, TrpA1 mediates ROS-induced Tk hormone release. In contrast, single-cell RNA-seq data[3] do not support *TrpA1* expression in AstA^+^ EECs, consistent with our findings that ROS does not promote AstA release – an effect that would be expected if TrpA1 were functionally expressed in AstA^+^ EECs. This contradicts the findings of Lin et al., who reported *TrpA1* expression in AstA^+^ EECs. We have now included relevant single-cell data in the revised manuscript (Fig. S6f) showing that *TrpA1* is specifically expressed in Tk^+^ EECs, but not in AstA^+^ EECs, and we have expanded the discussion to address discrepancies in TrpA1 expression and AstA regulation.

Taken together, our results reveal a dual-site regulatory mechanism in which Unpaired cytokines released from the gut act at the BBB to downregulate AstA receptor expression, while AstA release from EECs is simultaneously suppressed. We thank the reviewers for raising this important point. We have also included a discussion the other point raised by the reviewers – the possibility that ROS generated during sleep deprivation may engage the same signaling pathways described here, providing a mechanistic link between sleep deprivation, intestinal stress, and sleep regulation.

**Recommendations for the authors:**
A- Material and Methods:(1) Feeding Assay: The cited publication (doi.org:10.1371/journal.pone.0006063) states: "For the amount of label in the fly to reflect feeding, measurements must therefore be confined to the time period before label egestion commences, about 40 minutes in Drosophila, a time period during which disturbance of the flies affects their feeding behavior. There is thus a requirement for a method of measuring feeding in undisturbed conditions." Was blue fecal matter already present on the tube when flies were homogenized at 1 hour? If so, the assay may reflect gut capacity rather than food passage (as a proxy for food intake). In addition, was the variability of food intake among flies in the same tube tested (to make sure that 1-2 flies are a good proxy for the whole population)?

We agree that this is an important point for feeding experiments. We are aware of the methodological considerations highlighted in the cited study and have extensive experience using a range of feeding assays in *Drosophila*, including both short- and long-term consumption assays (e.g., dye-based and CAFE assays), as well as automated platforms such as FLIC and FlyPAD (Nature Communications, 2022; Nature Metabolism, 2022; and Nature Metabolism, 2025)[2,4,5].

For the dye-based assay, we carefully selected a 1-hour feeding window based on prior optimization. Since animals were not starved prior to the assay, shorter time points (e.g., 30 minutes) typically result in insufficient ingestion for reliable quantification. A 1-hour period provides a robust readout while remaining within the timeframe before significant label excretion occurs under our experimental conditions. To support the robustness of our findings, we complemented the dye-based assay with data from FLIC, which enables automated, high-resolution monitoring of feeding behavior in undisturbed animals over extended periods. The FLIC results were consistent with the dye-based data, strengthening our confidence in the conclusions. To minimize variability and ensure consistency across experiments, all feeding assays were performed at the same circadian time – Zeitgeber Time 0 (ZT0), corresponding to 10:00 AM when lights are turned on in our incubators. This time point coincides with the animals' natural morning feeding peak, allowing for reproducible comparisons across conditions. Regarding variability among flies within tubes, each biological replicate in the dye assay consisted of 1–2 flies, and results were averaged across multiple replicates. We observed good consistency across samples, suggesting that these small groups reliably reflect group-level feeding behavior under our conditions.

(2) Biological replicates: whereas the number of samples is clearly reported in each figure, the number of biological replicates is not indicated. Please include this information either in Material and methods or in the relevant figure legends. Please also include a description of what was considered a biological replicate.

We have now clarified in the Materials and Methods section under Statistics that all replicates represent independent biological samples, as suggested by the reviewers.

(3) Control Lines: please indicate which control lines were used instead of citing another publication. If preferred, this information could be supplied as a supplementary table.

We now provide a clear description of the control lines used in the Materials and Methods section. Specifically, all GAL4 and GAL80 lines used in this study were backcrossed for several generations into a shared *w*^1118^ background and then crossed to the same *w*^1118^ strain used as the genetic background for the *UAS-RNAi*, <i.CRISPR, or overexpression lines. This approach ensures, to a strong approximation, that the only difference between control and experimental animals is the presence or absence of the UAS transgene.

(4) Statistical analyses: for some results (e.g., those shown in Figure 3d), it could be useful to test the interaction between genotype and treatment.

We thank the reviewer for this helpful suggestion. In response, we have now performed two-way ANOVA analyses to assess genotype × treatment (diet) interaction effects for the relevant data, including those shown in Figure 3d as well as additional panels where animals were exposed to oxidative stress and sleep phenotypes were measured. We have added the corresponding interaction p-values in the updated figure legends for Figures 3d, 3k, 5a–c, 5f, 5h, 5i, 6c, 6e, and 7e. All of these tests revealed significant interaction effects, supporting the conclusion that the observed differences in sleep phenotypes are specifically dependent on the interaction between genetic manipulation (e.g., cytokine or receptor knockdown) and oxidative stress. These additions reinforce the interpretation that Unpaired cytokine signaling, glial JAK-STAT pathway activity, and AstA receptor regulation functionally interact with intestinal ROS exposure to modulate sleep. We thank the reviewer for suggesting this improvement.

(5) Reporting of p values. Some are reported as specific values whereas others are reported as less than a specific value. Please make this reporting consistent across different figures.

All p-values reported in the manuscript are exact, except in cases where values fall below p < 0.0001. In those instances, we use the inequality because the Prism software package (GraphPad, version 10), which was used for all statistical analyses, does not report more precise values. We believe this reporting approach reflects standard practice in the field.

(6) Please include the color code used in each figure, either in the figure itself or in the legend.

We have now clarified the color coding in all relevant figures. In particular, we acknowledge that the meaning of the half-colored circles used to indicate H_2_O_2_ treatment was not previously explained. These have now been clearly labeled in each figure to indicate treatment conditions.

(7) The scheme describing the experimental conditions and the associated chart is confusing. Please improve.

We have improved the schematic by replacing “ROS” with “H_2_O_2_” to more clearly indicate the experimental condition used. Additionally, we have added the corresponding circle annotations so that they now also appear consistently above the relevant charts. This revised layout enhances clarity and helps readers more easily interpret the experimental conditions. We believe these changes address the reviewer’s concern and make the figure significantly more intuitive.

1. Please indicate which line was used for upd-Gal4 and the evidence that it faithfully reflects upd3 expression.

We have now clarified in the Materials and Methods section that the upd3-*GAL4* line used in our study is Bloomington stock #98420, which drives GAL4 expression under the control of approximately 2 kb of sequence upstream of the *upd3* start codon. This line has previously been used as a transcriptional reporter for *upd3* activity. The only use of this line was to illustrate reporter expression in the EECs. To support this aspect of Upd3 expression, we now include new data in the revised manuscript using fluorescent in situ hybridization (FISH) against *upd3*, which confirms the presence of *upd3* transcripts in *prospero*-positive EECs of the adult midgut (Fig. S1b). Additionally, we show that *upd3* transcript levels are significantly reduced in dissected midguts following EEC-specific knockdown using multiple independent RNAi lines driven by *voilà-GAL4*, both alone and in combination with *R57C10-GAL80*, consistent with endogenous expression in these cells (Fig. 1a,b).

To further address the reviewer’s concern and provide additional support for the endogenous expression of *upd3* in EECs, we performed targeted knockdown experiments focusing on molecularly defined EEC subpopulations. The adult *Drosophila* midgut contains two major EEC subtypes characterized by their expression of Allatostatin C (AstC) or Tachykinin (Tk), which together encompass the vast majority of EECs. To selectively manipulate these populations, we used *AstC-GAL4* and *Tk-GAL4* drivers – both knock-in lines in which *GAL4* is inserted at the respective endogenous hormone loci. This design enables precise GAL4 expression in AstC- or Tk-expressing EECs based on their native transcriptional profile. To eliminate confounding neuronal expression, we combined these drivers with R57C10GAL80, restricting GAL4 activity to the gut and generating *AstCGut*> and *TkGut*> drivers. Using these tools, we knocked down *upd2* and *upd3* selectively in the AstC- or Tk-positive EECs. Knockdown of either cytokine in AstC-positive EECs significantly increased sleep under homeostatic conditions, recapitulating the phenotype observed with knockdown in all EECs (Fig. 1m-o). In contrast, knockdown of *upd2* or *upd3* in Tk-positive EECs had no effect on sleep (Fig. 1p-r). Furthermore, we show in the revised manuscript that selective knockdown of *upd2* or *upd3* in AstC-positive EECs abolishes the H_2_O_2_-induced increase in sleep (Fig. 3f–h). These findings demonstrate that Unpaired cytokine signaling from AstC-positive EECs is essential for mediating the sleep response to intestinal oxidative stress, highlighting this specific EEC subtype as a key source of cytokine-driven regulation in this context. These new results indicate that AstC-positive EECs are a primary source of the Unpaired cytokines that regulate sleep, while Tk-positive EECs do not appear to contribute to this function. Importantly, *upd3* transcript levels were significantly reduced in dissected midguts following *AstCGut* driven knockdown (Fig. S1r), further confirming that *upd3* is endogenously expressed in AstC-positive EECs. Thus we have bolstered our confidence that *upd3* is indeed expressed in EECs, as illustrated by the reporter line, through several means.

(9) Please indicate which GFP line was used with upd-Gal4 (CD8, NLS, un-tagged, etc). The Material and Methods section states that it was "UAS-mCD8::GFP (#5137);", however, the stain does not seem to match a cell membrane pattern but rather a nuclear or cytoplasmic pattern. This information would help the interpretation of Figure 1C.

We confirm that the GFP reporter line used with upd3-GAL4 was obtained from Bloomington stock #98420. As noted by the Bloomington *Drosophila* Stock Center, “the identity of the UAS-GFP transgene is a guess,” and the subcellular localization of the GFP fusion is therefore uncertain. We agree with the reviewer that the signal observed in Figure 1c does not display clear membrane localization and instead appears diffuse, consistent with cytoplasmic or partially nuclear localization. In any case, what we find most salient is the reporter’s labeling of Prospero-positive EECs in the adult midgut, consistent with *upd3* expression in these cells. This conclusion is further supported by multiple lines of evidence presented in the revised manuscript, as mentioned above in response to question #8: (1) fluorescent in situ hybridization (FISH) for *upd3* confirms expression in EECs (Fig. S1b), (2) EEC-specific RNAi knockdown of *upd3* reduces transcript levels in dissected midguts, and (3) publicly available single-cell RNA sequencing datasets[3] also indicate that *upd3* is expressed at low levels in a subset of adult midgut EECs under normal conditions. We have also clarified in the revised Materials and Methods section that GFP localization is undefined in the *upd3-GAL4* line, to guide interpretation of the reporter signal.

B- Results(1) Figure 1: According to previous work (10.1016/j.celrep.2015.06.009, http://flygutseq.buchonlab.com/data?gene=upd3%0D%0A), in basal conditions upd3 is expressed as following: ISC (35 RPKM), EB (98 RPKM), EC (57 RPKM), and EEC (8 RPKM). Accordingly, even complete KO in EECs should eliminate only a small fraction of upd3 from whole guts, even less considering the greater abundance of other cell types such as ECs compared to EECs. It would be useful to understand where this discrepancy comes from, in case it is affecting the conclusion of the manuscript. While this point per se does not affect the main conclusions of the manuscript, it makes the interpretation of the results more difficult.

We acknowledge the previously reported low expression of *upd3* in EECs. However, the FlyGut-seq site appears to be no longer available, so we could not directly compare other related genes. Nonetheless, our data – based on in situ hybridization, reporter expression, and multiple RNAi knockdowns – consistently support *upd3* expression in EECs. These complementary approaches strengthen the conclusion that EECs are an important source of systemic upd3 under the conditions tested.

(2) Figure 1: The upd2-3 mutants show sleep defects very similar to those of EEC>RNAi and >Cas9. It would thus be helpful to try to KO upd3 with other midgut drivers (An EC driver like Myo1A or 5966GS and a progenitor driver like Esg or 5961GS) to validate these results. Such experiments might identify precisely which cells are involved in the gut-brain signaling reported here.

We appreciate the reviewer’s suggestion and agree that exploring other potential sources of Upd3 in the gut is an interesting direction. In this study, we have focused on EECs, which are the primary hormone-secreting cells in the intestine and thus the most likely candidates for mediating systemic effects such as gut-to-brain signaling. While it is possible that other gut cell types – such as enterocytes (e.g., Myo1A^+^) or intestinal progenitors (e.g., Esg^+^) – also contribute to Upd3 production, these cells are not typically endocrine in nature. Demonstrating their involvement in gutto-brain communication would therefore require additional, extensive validation beyond the scope of the current study. Importantly, our data show that manipulating Upd3 specifically in EECs is both necessary and sufficient to modulate sleep in response to intestinal ROS, strongly supporting the conclusion that EEC-derived cytokine signaling underlies the observed phenotype. In contrast, manipulating cytokines in other gut cells could produce indirect effects – such as altered proliferation, epithelial integrity, or immune responses – that complicate the interpretation of behavioral outcomes like sleep. For these reasons, we chose to focus on EECs as the source of endocrine signals mediating gut-to-brain communication. However, to address this point raised by the reviewer, we have now included a statement in the Discussion acknowledging that other non-endocrine gut cell types may also contribute to the systemic Unpaired signaling that modulates sleep in response to intestinal oxidative stress.

(3) Figure 3: "This effect mirrored the upregulation observed with EEC-specific overexpression of upd3, indicating that it reflects physiologically relevant production of upd3 by the gut in response to oxidative stress." Please add (Figure 3a) at the end of this sentence.

We have now added “(Figure 3a)” at the end of the sentence to clearly reference the relevant data.

(4) For Figure 3b, do you have data showing that the increased amount of sleep was due to the addition of H2O2 per se, rather than the procedure of adding it?

We have added new data to address this point. To ensure that the observed sleep increase was specifically due to the presence of H_2_O_2_ and not an effect of the food replacement procedure, we performed a control experiment in which animals were fed standard food prepared using the same protocol and replaced daily, but without H_2_O_2_. These animals did not exhibit increased sleep, confirming that the sleep effect is attributable to intestinal ROS rather than the supplementation procedure itself (Fig. S3a). Thanks for the suggestion.

(5) In the text it is stated that "Since 1% H2O2 feeding induced robust responses both in upd3 expression and in sleep behavior, we asked whether gut-derived Unpaired signaling might be essential for the observed ROS-induced sleep modulation. Indeed, EEC-specific RNAi targeting upd2 or upd3 abolished the sleep response to 1% H2O2 feeding." While it is indeed true that there is no additional increase in sleep time due to EEC>upd3 RNAi, it is also true that EEC>upd3 RNAi flies, without any treatment, have already increased their sleep in the first place. It is then possible that rather than unpaired signaling being essential, an upper threshold for maximum sleep allowed by manipulation of these processes was reached. It would be useful to discuss this point.

Several findings argue against a ceiling effect and instead support a requirement for Unpaired signaling in mediating ROS-induced sleep. Animals with EEC-specific *upd2* or *upd3* knockdown or null mutation not only fail to increase sleep following H_2_O_2_ treatment but actually exhibit reduced sleep during oxidative stress (Fig. 3e, k, l; Fig. 5e, f), suggesting that Unpaired signaling is required to sustain sleep under these conditions. Similarly, animals with glial dome knockdown also show reduced sleep under oxidative stress, closely mirroring the phenotype of EEC-specific *upd3* RNAi animals (Fig. 5a–c, g–i). These results support the conclusion that gut-to-glia Unpaired cytokine signaling is necessary for maintaining elevated sleep during oxidative stress. In the absence of this signaling, animals exhibit increased wakefulness. We identify AstA as one such wake-promoting signal that is suppressed during intestinal stress. We present new data showing that this pathway is downregulated not only via Unpaired-JAK/STAT signaling in glial cells but also through reduced AstA release from the gut in the revised manuscript. This model, in which Unpaired cytokines promote sleep during intestinal stress by suppressing arousal pathways, is discussed throughout the manuscript to address the reviewer’s point.

(6) In Figure 3k, the dots highlighting the experiment show an empty profile, a full one, and a half one. Please define what the half dots represent.

We have now clarified the color coding in all relevant figures. Specifically, we acknowledge that the meaning of the half-colored circles indicating H_2_O_2_ treatment was not previously defined – it indicates washout or recovery time. In the revised version, these symbols are now clearly labeled in each figure to indicate the treatment condition, ensuring consistent and intuitive interpretation across all panels.

(7) The authors used appropriate GAL4 and RNAi lines to the knockdown dome, a upd2/3 JAK-STATlinked receptor, specifically in neurons and glia, respectively, in order to identify the CNS targets of upd2/3 cytokines produced by enteroendocrine cells (EECs). Pan-neuronal dome knockdown did not alter daytime sleep in adult females, yet pan-glial dome knockdown phenocopied effects of upd2/3 knockdown in EECs. They also observed that EEC-specific knockdown of upd2 and upd3 led to a decrease in JAK-STAT reporter activity in repo-positive glial cells. This supports the authors' conclusion that glial cells, not neurons, are the targets by which unpaired cytokines regulate sleep via JAK-STAT signaling. However, they do not show nighttime sleep data of pan-neuronal and pan-glial dome knockdowns. It would strengthen their conclusion if the nighttime sleep of pan-glial dome knockdown phenocopied the upd2/3 knockdowns as well, provided the pan-neuronal dome knockdown did not alter nighttime sleep.

We have now added nighttime sleep data for both pan-glial and pan-neuronal *domeless* knockdowns in the revised manuscript (Fig. 2a). Glial knockdown increased nighttime sleep, similar to EEC-specific *upd2/3* knockdown, while neuronal knockdown had no effect. These results further support the glial cells’ being the relevant target of gut-derived Unpaired signaling.

(8) The authors only used one method to induce oxidative stress (hydrogen peroxide feeding). It would strengthen their argument to test multiple methods of inducing oxidative stress, such as lipopolysaccharide (LPS) feeding. In addition, it would be useful to use a direct bacterial infection to confirm that in flies, the infection promotes sleep. Additionally, flies deficient in Dome in the BBB and infected should not be affected in their sleep by the infection. These experiments would provide direct support for the mechanism proposed. Finally, the authors should add a primary reference for using ROS as a model of bacterial infection and justify their choice better.

We agree that directly comparing different models of intestinal stress, such as bacterial infection or LPS feeding, would provide valuable insight into how gut-derived signals influence sleep in response to infection. As noted in our detailed responses above, we now include an expanded rationale for our use of H_2_O_2_ feeding as a controlled and well-established method for inducing intestinal ROS – one of the key physiological responses to enteric infection and inflammation. In the revised Discussion, we explicitly acknowledge that pathogenic infections – which trigger both intestinal ROS and additional immune pathways – may engage distinct or complementary mechanisms compared to chemically induced oxidative stress. We emphasize the importance of future studies aimed at dissecting these differences. In fact, we are actively pursuing this direction in ongoing work examining sleep responses to enteric infection. For the purposes of the present study, however, we chose to focus on a tractable and specific model of ROS-induced stress to define the contribution of Unpaired cytokine signaling to gut-brain communication and sleep regulation. This approach allowed us to isolate the effect of oxidative stress from other confounding immune stimuli and identify a glia-mediated signaling mechanism linking gut epithelial stress to changes in sleep behavior.

(9) To confirm that animals lacking EEC Unpaired signaling are not more susceptible to ROS-induced damage, the authors assessed the survival of upd2 and upd3 knockdowns on 1% H2O2 and concluded they display no additional sensitivity to oxidative stress compared to controls. It may be useful to include other tests of sensitivity to oxidative stress, in addition to survival.

We appreciate the reviewer’s suggestion. In our view, survival is a highly informative and stringent readout, as it reflects the overall physiological capacity of the animal to withstand oxidative stress. Importantly, our data show that animals lacking EEC-derived Unpaired signaling do not exhibit reduced survival following H_2_O_2_ exposure, indicating that their oxidative stress resistance is not compromised. Furthermore, we previously confirmed that feeding behavior is unaffected in these animals, suggesting that their ability to ingest food (and thus the stressor) is not impaired. As a molecular complement to these assays in response to this point and others, we have also performed an assessment of neuronal apoptosis (a TUNEL assay, Fig. S3f,g). This assay did not identify an increase in cell death in the brains of animals fed peroxide-containing medium. Thus, gross neurological health, behavior, and overall survival appear to be resilient to the environmental treatment regime we apply here, suggesting that the outcomes we observe arise from signaling per se.

(10) The authors confirmed that animals lacking EEC-derived upd3 displayed sleep suppression similar to controls in response to starvation. These results led the authors to conclude that there is a specific requirement for EEC-derived Unpaired signaling in responding to intestinal oxidative stress. However, they previously showed that EEC-specific knockdown of upd3 and upd2 led to increased daytime sleep under normal feeding conditions. Their interpretations of their data are inconsistent.

We appreciate the reviewer’s comment. While animals lacking EEC-derived Unpaired signaling show increased baseline sleep under normal feeding conditions, they still exhibit a robust reduction in sleep when subjected to starvation – comparable to that of control animals (Fig. S3h–j). This demonstrates that they retain the capacity to appropriately modulate sleep in response to metabolic stress. Thus, the sleep-promoting phenotype under normal conditions does not reflect a generalized inability to adjust sleep behavior. Rather, it highlights a specific role for Unpaired signaling in mediating sleep responses to intestinal oxidative stress, not in broadly regulating all sleep-modulating stimuli.

(11) The authors report a significant increase in JAK-STAT activity in surface glial cells at ZT0 in animals fed 1% H2O2-containing food for 20 hours. This response was abolished in animals with EECspecific knockdown of upd2 or upd3. The authors confirmed there were no unintended neuronal effects on upd2 or upd3 expression in the heads. They also observed an upregulation of dome transcript levels in the heads of animals with EEC-specific knockdown of upd3 fed 1% H2O2-containing food for 15 hours, which they interpret to be a compensatory mechanism in response to low levels of the ligand. This assay is inconsistent with previous experiments in which animals were fed hydrogen peroxide for 20 hours.

We thank the reviewer for identifying this discrepancy. The inconsistency arose from a labeling error in the manuscript. Both the JAK-STAT reporter assays in glial cells and the dome expression measurements were performed following 15 hours of H_2_O_2_ feeding, not 20 hours as previously stated. We have now corrected this in the revised manuscript.

(12) The authors show that animals with glia-specific dome knockdown did not have decreased survival on H2O2-containing food, and displayed normal rebound sleep in the morning following sleep deprivation. These results potentially undermine the significance of the paper. If the normal sleep response to oxidative stress is an important protective mechanism, why would oxidative stress not decrease survival in dome knockdown flies (that don't have the normal sleep response to oxidative stress)? This suggests that the proposed mechanism is not important for survival. The authors conclude that Dome-mediated JAK-STAT signaling in the glial cells specifically regulates ROS-induced sleep responses, which their results support.

We agree that our survival data show that glial dome knockdown does not reduce survival under continuous oxidative stress. However, we believe this does not undermine the importance of the sleep response as an adaptive mechanism. In our survival assay, animals were continuously exposed to 1% H_2_O_2_ without the opportunity to recover. In contrast, under natural conditions, oxidative stress is likely to be intermittent, and the ability to mount a sleep response may be particularly important for promoting recovery and maintaining homeostasis during or after transient stress episodes. Thus, while the JAK-STAT-mediated sleep response may not directly enhance survival under constant oxidative challenge, it likely plays a critical role in adaptive recovery under natural conditions.

(13) Altogether, the authors conclude that enteric oxidative stress induces the release of Unpaired cytokines which activate the JAK-STAT pathway in subperineurial glia of the BBB, which leads to the glial downregulation of receptors for AstA, which is a wake-promoting factor also released by EECs. This mechanism is supported by their results, however, this research raises some intriguing questions, such as the role of upd2 versus upd3, the role of AstA-R1 versus AstA-R2, the importance of this mechanism in terms of survival, the sex-specific nature of this mechanism, and the role that nutritional availability plays in the dual functionality of Unpaired cytokine signaling in regards to sleep.

We thank the reviewer for highlighting these important questions. Our data suggest that Upd2 and Upd3, while often considered partially redundant, both contribute to sleep regulation, with stronger effects observed for Upd3. This is consistent with prior studies indicating overlapping but non-identical roles for these cytokines. Similarly, although AstA-R1 and AstA-R2 can both be activated by AstA, knockdown of *AstA-R2* consistently produces more robust sleep phenotypes, suggesting a predominant role in mediating this effect. The possibility of sex-specific regulation is indeed compelling. While our study focused on females, many gut hormones show sex-dependent activity, and we recognize this as an important avenue for future research. Finally, we have included new data in the revised manuscript showing that gut-derived AstA is downregulated under oxidative stress, further supporting our model in which Unpaired signaling suppresses arousal pathways during intestinal stress

(14)Data Availability: It is indicated that: "Reasonable data requests will be fulfilled by the lead author". However, eLife's guidelines for data sharing require that all data associated with an article to be made freely and widely available.

We thank the reviewer for pointing this out. We have revised the Data Availability section of the manuscript to clarify that all data will be made freely available from the lead contact without restriction, in accordance with eLife’s open data policy.

References

(1) Li, Y., Zhou, X., Cheng, C., Ding, G., Zhao, P., Tan, K., Chen, L., Perrimon, N., Veenstra, J.A., Zhang, L., and Song, W. (2023). Gut AstA mediates sleep deprivaPon-induced energy wasPng in Drosophila. Cell Discov 9, 49. 10.1038/s41421-023-00541-3.(2)Ahrentlov, N., Kubrak, O., Lassen, M., Malita, A., Koyama, T., Frederiksen, A.S., Sigvardsen, C.M., John, A., Madsen, P., Halberg, K.A., et al. (2025). Protein-responsive gut hormone Tachykinin directs food choice and impacts lifespan. Nature Metabolism. 10.1038/s42255-025-01267-0.

(3) Li, H., Janssens, J., De Waegeneer, M., Kolluru, S.S., Davie, K., Gardeux, V., Saelens, W., David, F.P.A., Brbic, M., Spanier, K., et al. (2022). Fly Cell Atlas: A single-nucleus transcriptomic atlas of the adult fruit fly. Science 375, eabk2432. 10.1126/science.abk2432.

(4) Kubrak, O., Koyama, T., Ahrentlov, N., Jensen, L., Malita, A., Naseem, M.T., Lassen, M., Nagy, S., Texada, M.J., Halberg, K.V., and Rewitz, K. (2022). The gut hormone AllatostaPn C/SomatostaPn regulates food intake and metabolic homeostasis under nutrient stress. Nature communicaPons 13, 692. 10.1038/s41467-022-28268-x.

(5) Malita, A., Kubrak, O., Koyama, T., Ahrentlov, N., Texada, M.J., Nagy, S., Halberg, K.V., and Rewitz, K. (2022). A gut-derived hormone suppresses sugar appePte and regulates food choice in Drosophila. Nature Metabolism 4, 1532-1550. 10.1038/s42255-022-00672-z.